# A Neighborhood Search Integer Programming Approach for Wind Farm Layout Optimization

Juan-Andrés Pérez-Rúa[1], Mathias Stolpe[1], and Nicolaos Antonio Cutululis[1]

[1] Department of Wind and Energy Systems, Technical University of Denmark, Frederiksborgvej 399, 4000 Roskilde, Denmark

**Correspondence:** Juan-Andrés Pérez-Rúa (juru@dtu.dk)

**Abstract.** Two models and a heuristic algorithm to address the wind farm layout optimization problem are presented. The models are linear integer programming formulations where candidate locations of wind turbines are described by binary variables. One formulation considers an approximation of the power curve by means of a step-wise constant function. The other model is based on a power-curve-free model where minimization of a measure closely related to total wind speed deficit is optimized. A special-purpose neighborhood search heuristic wraps these formulations increasing tractability and effectiveness compared to the full model that is not contained in the heuristic. The heuristic iteratively searches neighborhoods around the incumbent using a branch-and-cut algorithm. The number of candidate locations and neighborhood sizes are adjusted adaptively. Numerical results on a set of publicly available benchmark problems indicate that a proxy for total wind speed deficit as objective is a functional approach, since high-quality solutions of annual energy production metric are obtained, when using the latter function as substitute objective. Furthermore, the proposed heuristic is able to provide good results compared to a large set of distinctive approaches that consider the turbine positions as continuous variables.

## 1  Introduction

### 1.1  Motivation and Problem Definition

Cost reductions for renewable energy generation is on the top of political agendas, with the objective of supporting the world-wide proliferation of clean energy production systems. Subsidy-free tendering processes become more frequent, as is the case for offshore wind auctions in Germany since 2017 and in Netherlands since 2018, or in China for onshore wind from 2021 (GWEC, 2020a). The fast evolution of offshore wind in the last decade, with a sharp growth of global installed capacity (GWEC, 2020b), is yet another clear indicator of growth trend of wind energy. For wind energy to become the cornerstone of a successful green energy transition, further reduction in costs - partly achievable by economically optimized wind farm designs - will play an important role.

The basic Wind Farm Layout Optimization (WFLO) problem aims at deciding the positioning of Wind Turbines (WTs) within a given project area to maximize the Annual Energy Production (AEP), while respecting a minimum separation distance. The classic problem definition aims at placing a fixed number $n_T$ of typically homogeneous (single type) WTs. This problem has been studied broadly and intensively since at least three decades (Herbert-Acero et al., 2014). The first effort in the topic

was the pioneering work of Mosetti et al. (1994), where the Katic-Jensen wake decay model (Katic et al., 1986), implemented to compute wake losses, is coupled with a genetic algorithm as optimizer to iteratively improve the layout.

## 1.2 Optimization Workflow for WFLO

The main components when building an optimization workflow for the WFLO problem are the wake models (deficit and superposition), the program formulation, and the associated numerical algorithms. For formulating tractable frameworks, the designer needs to rely on the so-called engineering wake models. These are essentially mathematical representations which can be expressed in terms of analytical equations after significantly simplifying complex physics modelling, while still capturing to a good extent the underlying nature of the phenomenon under analysis. Scientific articles in this field have proposed and validated engineering wake models with smooth and differentiable velocity deficit shape, two examples are Bastankhah's Gaussian (Bastankhah and Porté-Agel, 2016) or its simplified version (IEA Wind Task 37, 2019), and the Jensen cosine model (Jensen, N.O., 1983). Likewise, the aggregation of individual wake velocity deficits can be done through linear superposition (Lissaman, 1979) or root sum squares (Voutsinas et al., 1990), with local or freestream velocity conditions (Porté-Agel et al., 2020).

## 1.3 Continuous Optimization for WFLO

Optimization techniques for the WFLO problem formulation can be classified, depending on the choice of variables, into continuous and discrete optimization. In the field of continuous optimization, the location $\mathbf{p}_i$ of a WT $i$, in terms of the abscissa $(x_i)$ and ordinate variables $(y_i)$ in the Cartesian plane, $\mathbf{p}_i = (x_i, y_i)$, can take any real values, while ensuring that the point is within the project area $\mathbf{F}$, and simultaneously satisfying the minimum distance constraints. Several gradient-free algorithms have been applied to this problem, including metaheuristics, as genetic algorithm (Réthoré et al., 2014) or particle swarm optimization (Wan et al., 2010). Likewise, gradient-based methods can be used, as for example the Sparse Nonlinear OPTimizer (SNOPT), that uses a Sequential Quadratic Programming (SQP) approach (Thomas et al., 2022a), or interior-point solvers (Pérez et al., 2013). In general, metaheuristic algorithms, although highly flexible for modelling aspects, have considerably poorer scalability for larger problem sizes than gradient-based approaches (Stanley and Ning, 2019). Re-parametrization approaches aiming to reduce the number of variables through simplified geometrical representations of the problem, such as row and column spacing or inclination angle, are also emerging (Stanley and Ning, 2019). Additionally, multi-start strategies are frequently implemented as a workaround for the intrinsic multi-modal nature of the WFLO problem. Finally, hybrid methods combining gradient-free and gradient-based algorithms have been proposed with good results (Mittal and Mitra, 2017).

The utilization of simplified objective functions closely related to more sophisticated AEP models is also an emerging research field for continuous gradient-based optimization. In the recent work of LoCascio et al. (2022), a novel formulation for time-averaged wake velocity incorporating an analytical integral of wake deficits across wind direction is proposed. This article shows the application of this analytical formulation for WFLO using the Sequential Least Squares Quadratic Programming

(SLSQP) as numerical algorithm. Computational results indicate the ability of this approach in finding WT layouts with energy production comparable to the alternative of optimizing directly more accurate AEP objectives.

## 1.4 Discrete Optimization for WFLO

Discrete optimization models can be formulated for this problem by means of sampling the available project area in form of $N$ candidate location points. Thus, only a set of finite options from the continuous search space are considered, where the $n_T$ WTs to be installed are in principle $n_T \ll N$. In contrast to continuous optimization, a candidate point $i$ is then represented by a binary variable $\xi_i$, that gets a value of one if a WT is installed at that location, or zero otherwise. The vast majority of articles in the literature implement gradient-free algorithms for this technique, as the works of Mosetti et al. (1994) and

Grady et al. (2005), both using genetic algorithms. Algorithms utilizing explicit gradients are also a valid approach in this field (Pollini, 2022). This modelling technique fits very well in the well-studied general framework of integer programming. The main advantage of this approach is the possibility to utilize exact solvers based on branch-and-cut method; theoretically able to solve a problem to optimality while supporting common engineering constraints (Wolsey, 2020). Nevertheless, the low tractability and poor scalability of this method as function of the size of $N$ and the number of state variables is well-known.

A large number of benefits are implicit in the discrete modelling technique over the continuous counterpart, including: (i) capacity to include the number of WTs as a variable and to model overall economic metrics as Net Present Value (NPV) (Pollini, 2022), (ii) ease of modelling any shape of project area or forbidden zones, convex or non-convex, (iii) capacity to model extensive integrated models to support electrical systems optimization (Pérez-Rúa and Cutululis, 2022; Cazzaro et al., 2023), (iv) ease of modelling terrain-based constraints or cost functions (Cazzaro and Pisinger, 2022), (v) ease of incorporating

multiple WT types, among others. These functionalities are the main motivation for focusing on proposing new methods for the WFLO problem in the area of discrete optimization. Moreover, in broader terms, since even the basic definition of the WFLO problem translates into a non-convex formulation, new methods are required to efficiently obtain high-quality solutions.

## 1.5 Literature Review for Integer Programming within WFLO

Probably the first work within the context of integer programming for the WFLO problem was the thesis of Fagerfjäll in

2010 (Fagerfjäll, 2010), where a Mixed Integer Linear Program (MILP) is proposed, modelling the objective AEP function as a superposition of deficits defined in terms of power. Although physically inaccurate, as the deficit superposition should be computed for velocities, an important reduction in the number of variables is achieved that ultimately allow solving to optimality rather small problem instances. A similar approximation is carried out by Archer et al. (2011), Fischetti et al. (2016), and Quan and Kim (2019), but introducing important modifications to the model by reducing number of constraints.

The objective function may also be formulated for aggregated velocity deficit (Turner et al., 2014; Kuo et al., 2016), but the imperfect correspondence with AEP will result in not solving to optimality, possibly resulting in final low-quality solutions. Another advantage of integer programming formulations is the chance of incorporating heuristic routines in the top of such models, as for instance proximity search (Fischetti et al., 2016; Shaw, 1998), to quickly improve a given starting feasible point.

### 1.6 Contributions

Several contributions to the field of discrete optimization for WFLO are proposed in the manuscript. The first contribution is the proposition of new integer linear formulations which are able to capture to a good extent the underlying physics of the problem. The main obstacles for a MILP representation of WFLO problem are the non-linearity of the power curves, and the choice of wake velocity deficit superposition approach. Currently, the scientific literature has fundamental knowledge gaps. For example, as discussed before, previous works have considered aggregation of power deficits instead of velocities, gaining a simplification on the mathematical formulation in detriment of the physics modelling fidelity. This manuscript presents new strategies for modelling both facets in the class of MILP problems, one with explicit power curve and wake superposition modelling, and another with a proxy objective function based on total wind speed, thus simplifying the original formulation. In contrast to LoCascio et al. (2022), this proxy objective is developed for MILP optimization, meaning that the aim is to get a linear expression that does not need to be friendly for explicit gradient-based optimization.

The second main contribution is the proposition of a new special purpose neighborhood search heuristics in order to speed up the generation of high-quality solutions. This heuristic, wrapping both formulations, has a twofold functionality; first to increase tractability, and second to redirect the optimization search in terms of a specified objective function with higher fidelity. Similar neighborhood search methods have been proposed in the literature, as the Discrete exploration-based optimization (DEBO) (Thomas et al., 2022c), which is a two-steps process composed by a greedy initialization and a local search block. While the method proposed in this manuscript shares most of the advantages of the mentioned approach (no gradients required, can handle unconnected and non-convex boundary constraints, and so on), it actually goes beyond the DEBO algorithm as among others, i) significantly less AEP function evaluations are required, and ii) it is based on well-establish integer programming theory, relying in efficient implementations of the branch-and-cut algorithm. The main numerical results indicate good computational performances for a set of publicly available benchmark case studies compared to state-of-the-art gradient-free and gradient-based approaches (Baker et al., 2019).

The rest of the manuscript is structured as follows. Section 2 introduces the engineering models of the physical aspects of interest. Section 3 presents the two mathematical programs developed, and Sect. 4 describes the proposed heuristic framework wrapping both programs. Computational experiments are shown in Sect. 5, followed up by discussions in Sect. 6, and lastly the manuscript is finalized with the conclusions in Sect. 7.

## 2 Physics Modelling

The proposed MILP models and general optimization framework in this manuscript can be easily applied to many wake deficit models. No particular properties on smoothness or differentiability are required from these models for optimization purposes. Additionally, no specific demands on mathematical structure in connection with controlling wake diameter and deficit (Thomas et al., 2022b) are stemming from the optimization programs proposed in this article. Since the computational results in the article are obtained after solving open access case studies from the IEA Wind Task 37 (Baker et al., 2019), the wake model implemented there is presented in Sect. 2.1, along with the superposition techniques in Sect. 2.2, WT power curve in Sect.

2.3, and the AEP calculation procedure in Sect. 2.4. Variations on ways of computing the absolute velocity deficits and linear wakes superposition under the framework of MILP are also introduced.

## 2.1 Wake Deficit Model

A simplified version of Bastankhah's Gaussian is considered (IEA Wind Task 37, 2019). The relative velocity deficit $\delta_{i\ell} = \Delta_{i\ell}/u_\infty = (u_\infty - u(\bar{x}_i, \bar{y}_i))/u_\infty$ behind a single WT located at $\ell$, and evaluated at point $i$, is described using the model and notation from Case Study I (IEA Wind Task 37, 2019).

$$\delta_{i\ell} = \begin{cases} \left(1 - \sqrt{1 - \frac{C_\mathrm{T}}{8\sigma_\mathrm{y}^2/D^2}}\right) \exp\left(-0.5\left(\frac{\bar{y}_i - \bar{y}_\ell}{\sigma_\mathrm{y}}\right)^2\right), & \bar{x}_i - \bar{x}_\ell > 0 \\ 0, & \text{otherwise.} \end{cases} \tag{1}$$

$$\sigma_\mathrm{y} = k_\mathrm{y}(\bar{x}_i - \bar{x}_\ell) + D/\sqrt{8} \tag{2}$$

where $u_\infty$ is the inflow wind speed, $C_\mathrm{T}$ is the thrust coefficient, $\bar{x}_i - \bar{x}_\ell$ is the stream-wise distance from the hub generating wake ($\bar{x}_\ell$) to hub of interest ($\bar{x}_i$) along freestream (let this difference be $d_{i\ell}^{\parallel}$), $\bar{y}_i - \bar{y}_\ell$ is the span-wise distance from the hub generating wake to hub of interest perpendicular to freestream (let this difference be $d_{i\ell}^{\perp}$), $\sigma_\mathrm{y}$ is the standard deviation of the wake deficit, $k_\mathrm{y}$ is a variable based on a turbulence intensity, and $D$ is the WT diameter.

## 2.2 Wake Velocity Deficit Superposition Model

The absolute velocity deficit $\Delta_{i\ell}(\theta^j, k)$ at wind direction $\theta^j$ and wind speed index $k$ can be estimated in two ways. Either based on the inflow wind speed (Lissaman, 1979; Katic et al., 1986) through

$$\Delta_{i\ell}(\theta^j, k) = \delta_{i\ell}(\theta^j, k)u_\infty^k \tag{3}$$

or based on the wind speed $u_{\ell jk}$ at WT $\ell$ creating the wake at point $i$ for wind direction $\theta^j$ and speed $k$ (Voutsinas et al., 1990; Niayifar and Porté-Agel, 2015),

$$\Delta_{i\ell}(\theta^j, k) = \delta_{i\ell}(\theta^j, k)u_{\ell jk} \tag{4}$$

here $\delta_{i\ell}(\theta^j, k)$ is the relative velocity deficit of $\ell$ over $i$ at operation condition $\{j, k\}$ after Eq.(1) and Eq.(2). Note that Eq. (3) leads to a greater value and therefore is considered a conservative approach compared to (the potentially more realistic) Eq. (4). Nonetheless, implementing Eq. (3) greatly simplifies the resultant system of equations and allow for preprocessing calculations.

Let the set $\boldsymbol{U}_i^{\theta^j}$ collect the WTs creating wake over WT at point $i$ for wind direction $\theta^j$ as per

$$\boldsymbol{U}_i^{\theta^j} = \{\ell \mid \text{position } \ell \text{ is up-wind compared to position } i \text{ for wind direction } j\} \tag{5}$$

The wake velocity deficit superposition, to calculate the total velocity deficit at WT $i$, $\Delta_i(\theta^j, k)$, can be obtained through two mechanisms. Either it is based on linear superposition model (Lissaman, 1979; Niayifar and Porté-Agel, 2015) through

$$\Delta_i(\theta^j, k) = \sum_{\ell \in \boldsymbol{U}_i^{\theta j}} \Delta_{i\ell}(\theta^j, k) \tag{6}$$

or it is based on the root sum squares superposition model (Katic et al., 1986; Voutsinas et al., 1990)

$$\Delta_i(\theta^j, k) = \sqrt{\sum_{\ell \in \boldsymbol{U}_i^{\theta j}} \Delta_{i\ell}^2(\theta^j, k)} \tag{7}$$

## 2.3 WT Power Curve

Suitable power curves are required for computing AEP. Often, power curves are not perfectly suitable for optimization, due to the usual non-differentiability in several points throughout the function. Generally, a power curve is zero below cut-in wind speed, zero above the cut-out wind speed, and constant between the rated wind speed and the cut-out wind speed. In this particular study, between the cut-in and rated wind speeds the curve is assumed to be smooth, convex and monotonically increasing. The simplified power curve for a generic turbine as a function of wind speed $u$ is modelled through

$$p(u) = \begin{cases} 0, & u < u^{\text{cut-in}} \\ p^{\text{rated}} \left( \frac{u - u^{\text{cut-in}}}{u^{\text{rated}} - u^{\text{cut-in}}} \right)^3, & u^{\text{cut-in}} \leq u < u^{\text{rated}} \\ p^{\text{rated}}, & u^{\text{rated}} \leq u < u^{\text{cut-out}} \\ 0, & u \geq u^{\text{cut-out}}. \end{cases} \tag{8}$$

where $p^{\text{rated}}$ is the nominal power at (and above) rated wind speed $u^{\text{rated}}$. The other turbine characteristics are the cut-in wind speed $u^{\text{cut-in}}$, and the cut-out wind speed $u^{\text{cut-out}}$. In this definition, the WT power curve is not differentiable at $u^{\text{cut-in}}$, $u^{\text{rated}}$, $u^{\text{cut-out}}$, since in these points the left and right hand side derivatives are different. Be aware that the optimization programs proposed in this manuscript are not dependent on WT power curve differentiability.

## 2.4 Annual Energy Production, AEP

The AEP is calculated with

$$AEP = 8760 \sum_{i=1}^{n_{\text{T}}} \sum_{j,k} w_{jk} p(u_{ijk}) \tag{9}$$

where $w_{jk}$ is the joint probability of wind direction $j$ and wind speed $k$, and 8760 is the number of hours of a standard year.

## 3 Optimization Models

The MILP program with explicit modelling of the WT power curve, wake deficit, and wakes superposition, is introduced in Sect. 3.1. Then, the power-curve-free formulation is described in Sect. 3.2.

The main type of variables $\xi_i \in \{0,1\}$ represent presence or absence of turbines at the candidate locations, for both models. Given $N$ points, i.e. candidate locations for turbine positions, with positions $\mathbf{p}_i$ inside the domain $\mathbf{F}$ (i.e. $\mathbf{p}_i \in \mathbf{F}$ for all $i$ WT candidate locations), binary variables $\xi_i \in \{0,1\}$ are associated with the following interpretation

$$\xi_i = \begin{cases} 1, & \text{if a turbine is located at point } i \text{ with position } \mathbf{p}_i, \text{ and} \\ 0, & \text{otherwise.} \end{cases} \tag{10}$$

Let the index sets $\mathbf{N}_i$ storing the candidate locations violating the minimum distance constraints for a WT $i$ be defined as

$$\mathbf{N}_i = \{q \in \{1,\ldots,N\}, q \neq i \mid d_{iq}(\mathbf{p}_i,\mathbf{p}_q) < d^{\min}\} \tag{11}$$

where $d^{\min} > 0$ is the minimum required distance between two turbines. If $\xi_i = 1$ then all binary variables in the set $\mathbf{N}_i$ should be forced to zero, whereas if $\xi_i = 0$ these variables should be free to take any value in $\{0,1\}$.

All relevant distances can be pre-processed for all combinations of points $i$ and $q$. These parameters are then defined as
function of the Cartesian plane positions $\mathbf{p}$ and wind direction $\theta^j$, as the Euclidean distances $d_{iq}(\mathbf{p}) = \|\mathbf{p}_i - \mathbf{p}_q\|_2$, the stream-wise distances $d_{iq}^{\parallel}(\mathbf{p};\theta^j)$ and the span-wise distances $d_{iq}^{\perp}(\mathbf{p};\theta^j)$, extending the concept introduced in Sect. 2.1.

### 3.1   Power-curve-based Model

Continuous state variables $u_{ijk}$ are used for wake modelling and power computation. A variable $u_{ijk}$ represents the wind speed at WT location $i$, for wind direction $j$, and wind speed $k$.

The power curve is approximated with a step-wise function. The cubic part of the power curve is first partitioned into $m$ intervals, plus one interval from a negative point $(-u^{\text{ini}})$ to the cut-in speed, and a final one to cover the range from rated to cut-out speed. Each isometric interval within the cubic domain of length $\Delta u = (u^{\text{rated}} - u^{\text{cut-in}})/m$, is approximated with a constant power value, see Fig. 1.

An interval $l$ of the whole domain is characterized by three parameters $u_{\text{s}}^l$, $u_{\text{m}}^l$, and $u_{\text{h}}^l$ with the next properties

$$u_{\text{s}}^1 = -u^{\text{ini}}, u_{\text{h}}^1 = u^{\text{cut-in}}, u_{\text{s}}^{m+2} = u^{\text{rated}}, u_{\text{h}}^{m+2} = u^{\text{cut-out}} \tag{12}$$

$$u_{\text{s}}^2 = u^{\text{cut-in}}, u_{\text{h}}^{m+1} = u^{\text{rated}} \tag{13}$$

$$u_{\text{s}}^{a+1} = u^{\text{cut-in}} + (a-1)\Delta u \text{ for } a = 1,\ldots,m \tag{14}$$

$$u_{\text{h}}^{a+1} = u^{\text{cut-in}} + a\Delta u \text{ for } a = 1,\ldots,m \tag{15}$$

$$u_{\text{m}}^l = 0.5(u_{\text{s}}^l + u_{\text{h}}^l) \tag{16}$$

equation (12) defines the lower and upper limits for the extreme intervals $l = 1$ and $l = m + 2$, Eq. (13) formalizes the lower and upper limits for the first interval in the cubic part, $a = 1$, and the last one $a = m$, respectively. Equation (14) expresses the lower limits for intervals in the cubic part ($a = 1,\ldots,m$), while Eq. (15) does it for the upper limits. Equation (16) presents how to determine the extracted wind speed associated to the interval $l$ of within whole domain, which is the average value of $u_{\text{s}}^l$ and $u_{\text{h}}^l$.

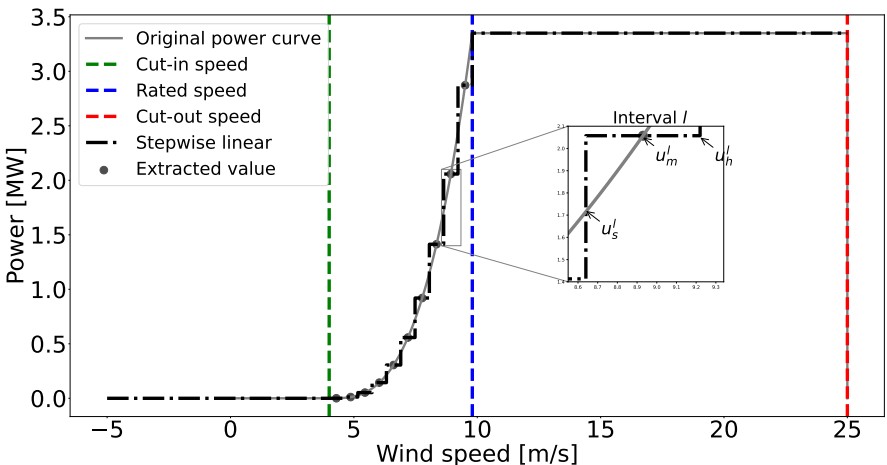

**Figure 1.** Piece-wise constant approximation of a wind turbine power curve through sampling with $m = 10$ intervals between the cut-in and rated wind speeds.

Let binary state variables $\eta_{ijk}^l \in \{0,1\}$ for $l = 1, \ldots, m+2$ be defined with the interpretation

$$\eta_{ijk}^l = \begin{cases} 1, & \text{if } u_s^l \leq u_{ijk} \leq u_h^l, \text{ and} \\ 0, & \text{otherwise.} \end{cases} \tag{17}$$

i.e. these variables indicate which of the wind speed intervals $l$ of the power curve approximation for WT $i$, operates at wind direction $j$, and wind speed $k$.

With all the variables of the model - activation variables $\xi$, continuous state variables $u$, and binary state variables $\eta$ - introduced, formulation in Eq. (18) follows,

$$\underset{\xi,\eta,u}{\text{maximize}} \; 8760 \sum_{i=1}^{N} \sum_{j,k} \sum_{l=1}^{m+2} w_{jk} \eta_{ijk}^l p(u_m^l) \tag{18a}$$

$$\text{subject to: } \xi_i + \xi_q \leq 1 \qquad\qquad \forall\, i,q \in \boldsymbol{N}_i \tag{18b}$$

$$n^{\min} \leq \sum_{i=1}^{N} \xi_j \leq n^{\max} \tag{18c}$$

$$\sum_{l=1}^{m+2} \eta_{ijk}^l u_s^l \leq u_{ijk} \leq \sum_{l=1}^{m+2} \eta_{ijk}^l u_h^l \qquad\qquad \forall\, (i,j,k) \tag{18d}$$

$$\sum_{l=1}^{m+2} \eta_{ijk}^l = 1 \qquad\qquad \forall\, (i,j,k) \tag{18e}$$

$$u_{ijk} = u_\infty^k \left( \xi_i - \sum_{\ell \in \boldsymbol{U}_i^{\theta^j}} \xi_\ell \delta_{i\ell}(\theta^j, u_\infty^k) \right) \qquad\qquad \forall\, (i,j,k) \tag{18f}$$

$$\xi, \eta \in \{0,1\} \quad u \in \mathbb{R} \tag{18g}$$

This program collects the AEP objective function, the constraints of a generalized version of the WFLO problem, and the variables' domain definition. The objective function in Eq. (18a) is an approximation of the AEP computation presented in Eq. (9). Equation (18b) models the minimum distance constraints as explained in the introduction of Sect. 3. If a binary variable $\xi_i$ is active, then all candidate points closer than $d^{\min}$ should be excluded, i.e. set to zero. If a binary variable $\xi_i$ is inactive then the other candidates are still eligible. The definition of the set $\boldsymbol{N}_i$ is provided in Eq. (11). Equation (18c) models the situation that the designer requires at least $n^{\min}$ and at most $n^{\max}$ WTs to be located in the domain. Note that for the classic problem definition $n^{\min} = n^{\max} = n_{\mathrm{T}}$. Equation (18d) connects state variables $u$ and $\eta$ as explained in Eq. (17) while Eq. (18e) forces one operation case active for each WT candidate at each wind direction and speed. The last constraint in Eq. (18f) is for the wake velocity deficit and wakes superposition modelling to calculate wind speed for each candidate location at each wind direction and inflow speed $u_\infty^k$. The presented model supports a conservative velocity deficit approach (Eq. (3)) with linear superposition (Eq. (6)). The definition of set $\boldsymbol{U}_i^{\theta^j}$ is provided in Eq. (5). Note that an extension, consisting in creating extra continuous state variables and associated constraints, could allow for considering the more realistic approach in Eq. (4). It is still unknown if the root sum squares model of Eq. (7) could be implemented in the framework of MILP. Finally, Eq. (18g) defines the domain of the required variables. A value for $u^{\mathrm{ini}}$ of $u^{\mathrm{cut\text{-}out}}$ is set up.

## 3.2 Power-curve-free Model

Albeit the formulation of Sect. 3.1 represents to a very large extent the physics ruling the problem, it has a considerable number of variables and constraints that may hinder the capacity to tackle larger problems. The model presented in this section neglects the power curve and AEP calculation and aims at simplifying the power-curve-based version.

The power-curve-free model introduces a strategy to account for the combination of Eq. (3) and Eq. (7) to calculate velocities, since the case studies from the IEA Wind Task 37 follow this methodology for AEP computation. It would be possible though to consider the linear superposition model if necessary. However, the power-curve-free model does not support the application of Eq. (4).

Combining Eq. (3) and Eq. (7) and extending the summation range in Eq. (7) to all candidate locations, the sum of wind speeds in the farm, $U$, can be modelled through

$$U = \sum_{i=1}^{N} \sum_{j,k} w_{jk} u_\infty^k \xi_i - \sum_{i=1}^{N} \sum_{j,k} w_{jk} u_\infty^k \sqrt{\sum_{\ell=1}^{N} (\delta_{i\ell}(\theta^j, u_\infty^k))^2 z_{i\ell}} \tag{19}$$

where new binary variables $z_{i\ell}$ are introduced. The variable $z_{i\ell}$ is equal to one if both WTs $i$ and $\ell$ are active (i.e. if $\xi_i = \xi_\ell = 1$) and zero otherwise. Nevertheless, the previous expression is not linear for variable $z_{i\ell}$ due to the presence of the square root in each total relative velocity deficit term. By removing the square roots, the following expression is obtained:

$$\tilde{U} = \overbrace{\sum_{i=1}^{N} \sum_{j,k} w_{jk} u_\infty^k \xi_i}^{\text{Total inflow wind speed}} - \overbrace{\sum_{i=1}^{N} \sum_{\ell=1}^{N} \sum_{j,k} w_{jk} u_\infty^k (\delta_{i\ell}(\theta^j, u_\infty^k))^2 z_{i\ell}}^{\text{Total wind speed deficit proxy}} \tag{20}$$

the arguments of the square roots in Eq. (19) define a function closely related to the full root-squared expression. This linearization approach is similar to the one proposed by Turner et al. (2014). Let the pre-processed coefficient in front of of $z_{i\ell}$ be

$$b_{i\ell} = \sum_{j,k} w_{jk} u_\infty^k (\delta_{i\ell}(\theta^j, u_\infty^k))^2 \tag{21}$$

combining Eq. (20) and Eq. (21) results in

$$\tilde{U} = \overbrace{\sum_{i=1}^{N} \sum_{j,k} w_{jk} u_\infty^k \xi_i}^{\text{Total inflow wind speed}} - \overbrace{\sum_{i=1}^{N} \sum_{\ell>i}^{N} (b_{i\ell} + b_{\ell i}) z_{i\ell}}^{\text{Total wind speed deficit proxy}} \tag{22}$$

which defines the objective function of the power-curve-free model. In comparison to the objective function in Eq. (18a), no power curve or continuous state variables are required.

Nonetheless, the presence of variables $z_{i\ell}$ can be troublesome. For the complete model, in addition to having these variables of combinatorial nature, constraints of the same kind must be incorporated: $z_{ij} \geq \xi_i + \xi_j - 1$, $z_{ij} \leq \xi_i$, $z_{ij} \leq \xi_j$, $z_{ij} \geq 0$. Experimental results show the heavy computational burden incurred when solving this formulation, impacting the ability of solving large-scale problems (Fischetti et al., 2016). To circumvent this, a big-M trick is incorporated (Wolsey, 2020), resulting in an exactly equivalent model, as reflected in formulation of Eq. (23).

The new objective function in Eq. (23a) modifies the component linked to the total wind speed deficit proxy by creating variables $\tau_i$; this variable means total wind speed deficit proxy for WT in candidate location $i$. Equation (23b) defines $\tau_i$, if a WT candidate location is inactive $\xi_i = 0$, then there is no deficit at this location, therefore $\tau_i = 0$, because of $M_i = \sum_{\ell=1:i\neq\ell}^{N} b_{i\ell}$, and the minimization nature of the problem for wind speed deficits. Oppositely, if $\xi_i = 1$, then $\tau_i$ is forced to be equal to $\sum_{\ell=1:i\neq\ell}^{N} \xi_\ell b_{i\ell}$. The next two equations are the same with those already presented in Sect. 3.1 for number of active WTs, and minimum distance constraints. Finally, Eq. (23e) defines the domain of the required variables.

$$\underset{\xi,\tau}{\text{maximize}} \overbrace{\sum_{i=1}^{N} \sum_{j,k} w_{jk} u_\infty^k \xi_i}^{\text{Total inflow wind speed}} - \overbrace{\sum_{i=1}^{N} \tau_i}^{\text{Total wind speed deficit proxy}} \tag{23a}$$

$$\text{subject to: } \tau_i \geq \sum_{\ell=1:i\neq\ell}^{N} \xi_\ell b_{i\ell} + (\xi_i - 1) M_i \qquad \forall\, i \tag{23b}$$

$$n^{\min} \leq \sum_{i=1}^{N} \xi_i \leq n^{\max} \tag{23c}$$

$$\xi_i + \xi_q \leq 1 \qquad \forall\, i, q \in \boldsymbol{N}_i \tag{23d}$$

$$\xi \in \{0,1\} \quad \tau \in \mathbb{R} : \tau \geq 0 \tag{23e}$$

note that for the classic problem definition $n^{\min} = n^{\max} = n_{\mathrm{T}}$, the first part of the objective function becomes

$$\sum_{i=1}^{N} \sum_{j,k} w_{jk} u_\infty^k \xi_i = \sum_{j,k} w_{jk} u_\infty^k \sum_{i=1}^{N} \xi_i = \sum_{j,k} w_{jk} u_\infty^k n_{\mathrm{T}} = \text{constant}$$

for this situation, the objective function is thus equivalent to

$$\underset{\xi,\tau}{\text{minimize}} \sum_{i=1}^{N} \tau_i \tag{24}$$

this proxy objective function is very useful for formulating the program in the MILP category. While the work by LoCascio et al. (2022) focuses on a different formulation (likely more accurate analytically than the one presented here) that is non-linear but gradient friendly, hence useful for continuous gradient-based optimization.

Compared to Turner et al. (2014), the MILP program (23) with objective replaced by Eq. (24), linearizes the complexity of its largest set of constraints and variables from $N^2$ to $N$ (Eq. (23b) and Eq. (23e)). Furthermore, the constraints in Eq. (23d), which can lead to infeasible points, are not neglected as by Turner et al. (2014).

## 4  Neighborhood Search Heuristic

For addressing large-scale problems, a heuristic wrapping the MILP formulations given in Sect. 3 is introduced. It is based on neighborhood search and local branching theory (Fischetti and Lodi, 2003). The algorithm solves a sequence of MILPs, with different candidates number $N$ and/or neighborhood search size $K$, taking advantage of robust and efficient implementations of branch-and-cut methods for MILP. The heuristic relies on the observation that for a fixed layout described by $\xi \in \{0,1\}^N$, the other state variables are straightforward to determine. This observation is valid for all problem formulations presented in Sect. 3. Given $\xi \in \{0,1\}^N$, for the power-curve-based model, the value of continuous state variables $u$ can be found through classical wake analysis, and the binary state variables $\eta$ are directly determined by inspection of the velocities. Similarly, for the power-curve-free model, the $\tau$ variables are trivially computed. The pseudo code of the Neighborhood Search Heuristic (NSH) is described in detail in Algorithm 1.

The first three lines are the main inputs of the algorithm: the candidates set $C$, the times set $T$, and neighborhood sizes set $V$. The first set contains the sizes $N$ of the meshes to be considered, the second one is the maximum computing time $T$ for the MILP solver for each size $N$ and the last one is for the search size defined as the maximum number of changes $K$ allowed to the incumbent. If the incumbent is improved, then the candidates set $C$, and neighborhood size $K$ are kept, otherwise at least one of them is increased. The first step (line 5) is to obtain an initial incumbent binary variables, with the set $\xi$ storing the acquired value (0 or 1) for each variable $\xi_i : i \leq N$. The incumbent has an objective value of $o_b$ calculated after the ***true objective function***. The true objective function refers to the real equation that represents the ultimate aim to be optimized. For example, if this is the AEP, then it is the product of the power calculation process, applying the considered wake and superposition models and the original power curve, and not the objective function of the implemented formulation, as in Eq. (18a), which is always an approximation.

The next step is to start the iterative process in line 6. Values for $N$, $T$, and $K$ are fetched in line 7, followed by the formulation of the MILP model for candidates $N$ accounting for the active locations in $\xi$. The Hamming distance, see e.g. Fischetti and Lodi (2003), centered around the incumbent point $\xi$, is added to the optimization model in line 9; this constraint

reduces the search space as the number of changes of $\xi$ are limited to $K$. The complete model is sent to the MILP solver with $\xi$ as warm-starter, stopped when reaches either optimality or the assigned maximum computing time $T$.

---

**Algorithm 1** Neighborhood Search Heuristic (NSH) Algorithm

---

1: $\boldsymbol{C} \leftarrow \{N_1, \cdots, N_C\}, N \in \boldsymbol{C}$           {**Input candidates set**}

2: $\boldsymbol{T} \leftarrow \{T_1, \cdots, T_C\}, T \in \boldsymbol{T}$           {**Input times set**}

3: $\boldsymbol{V} \leftarrow \{K_1, \cdots, K_V\}, K \in \boldsymbol{V}$          {**Input neighborhood sizes set**}

4: $countern \leftarrow 1 \quad counterv \leftarrow 1$

5: *Obtain initial incumbent of activation binary variables for WTs $\xi$ with objective value $o_b$*

6: **for** $(\kappa = 1 : 1 : \kappa_{max})$ **do**

7:     $N \leftarrow \boldsymbol{C}[countern] \quad T \leftarrow \boldsymbol{T}[countern] \quad K \leftarrow \boldsymbol{V}[counterv]$

8:     *Formulate optimization model with $N$ candidates (including the incumbent), either from Sect. 3.1 or Sect. 3.2*

9:     *Add Hamming distance constraint centered around the incumbent $\xi$, $\sum_{i:\xi_i=0} \xi_i + \sum_{i:\xi_i=1}(1-\xi_i) \leq K$*

10:     *Solve opt. model from algorithm lines (8) to (9) until optimality or computing time $T$ with $\xi$ as warm-starter*

11:     *Get the solution pool $\mathbf{S}$, where $\hat{\xi} \in \mathbf{S}$ represents the activation binary variables for WTs of an individual point*

12:     *Apply **true objective function** over each point $\hat{\xi} \in \mathbf{S}$, and obtain objective values set $\boldsymbol{O}$*

13:     *Compute $o_t \leftarrow \max \boldsymbol{O}$, and $i_t \leftarrow \arg\max \boldsymbol{O}$*

14:     **if** $o_t > o_b$ **then**

15:        $o_b \leftarrow o_t$

16:        $\xi \leftarrow \mathbf{S}[i_t]$

17:     **else**

18:        $counterv \leftarrow counterv + 1$

19:     **end if**

20:     **if** $counterv = |\boldsymbol{V}| + 1$ **then**

21:        $counterv \leftarrow 1 \quad countern \leftarrow countern + 1$

22:     **end if**

23:     **if** $countern = |\boldsymbol{C}| + 1$ **then**

24:        *Break*

25:     **end if**

26: **end for**

---

After solver termination, the solution pool $\mathbf{S}$ is retrieved in line 11. The solution pool contains all the feasible layouts obtained in an iteration $\kappa$ from the MILP solver. These points are a result of a linear programming relaxation or from applying heuristics in a given node, such as, relaxation induced search, polishing, and feasibility pump (IBM, 2022). It is very important to emphasize the aim of getting the whole pool instead of the best point. This is done because of the imperfect correspondence between the true objective function and the objective function of the applied MILP model. For example, a solution which

may have worse objective value, may actually have a better AEP based on the real model. One of the advantages of the NSH compared to the DEBO algorithm by Thomas et al. (2022c) is the reduced number of AEP evaluations. In an iteration $\kappa$, only $|\mathbf{S}|$ evaluations are required. Likewise, many of the other expensive calculations are done in a pre-processing stage. The whole pool of solutions is examined, and the best solution indexed by $i_t$ with AEP of $o_t$ is obtained in line 13. If $o_t$ is actually greater than $o_b$, then the whole algorithm is re-centered around the new $\xi$ (lines 14 to 16) and in the next iteration $\kappa$, the same values of $N$ and $K$ are maintained. Otherwise, the next value of $K$ is taken (line 18), unless the set has been exhausted. In this case, the next candidates size $N$ is considered given by $countern$, restarting the neighborhood set counter $counterv$ to one (lines 20 to 22). The NSH algorithm is terminated when all candidates set $C$ have been processed (line 23 to 25). Another difference between the NSH and the DEBO is that the latter only changes the position of a single WT in a given iteration, while the former considers simultaneous modifications of several WT positions.

## 5   Computational Experiments

For a transparent benchmark of the proposed methods, the open access case studies from the IEA Wind Task 37 in Baker et al. (2019) are used for comparison. The Task 37 cases consider circular project areas with three different radius (1300 m, 2000 m, and 3000 m) and number of WTs (16, 36, and 64), $n_T$. Thus, Case I has a radius of 1300 m and $n_T = 16$ WTs, whereas Case II has radius 2000 m and $n_T = 36$, and Case III has radius 3000 m and $n_T = 64$, correspondingly.

The results of the statistical correlation between the proxy function given by the argument in Eq. (24) and AEP of the problem definition of Baker et al. (2019) are presented in Sect. 5.1 for each case. The performance of the proposed models in the case studies are shown in Sects. 5.2 (Case I), 5.3 (Case II), 5.4 (Case III). The power-curve-free model of Eq. (23) is implemented with the Eq. (24) as objective function in these three sections. The true objective function in the NSH Algorithm 1 for these cases is the AEP of the problem definition. In the end, to prove the capabilities of power-curve-based model of Eq. (18), Sect. 5.5 displays results after applying this formulation with a modified objective function to express a metric similar to NPV.

The main parameters of the wake model in Sect. 2.1 are fixed to $C_T = 8/9$ and $k_y = 0.0324555$, according to Baker et al. (2019). The wind resource is modelled using a wind rose approach where the wind resource is binned in $J$ directions, and for a specific direction $j$ $(\theta^j)$, wind speeds are discretized in $\Upsilon$ sectors. For the case studies, the wind rose is composed of 16 directions and a single wind speed $k$ of 9.8 ms$^{-1}$, shown in Fig. 2. The power curve from Eq. (8) modelling the IEA37 3.35 MW reference turbine (with diameter of $D = 130$ m) is used in the case studies, ensuring replicability of results (IEA Wind Task 37, 2019; Baker et al., 2019). The main parameters are $p^{\text{rated}} = 3.35$ MW, $u^{\text{rated}} = 9.8$ ms$^{-1}$, $u^{\text{cut-in}} = 4$ ms$^{-1}$, and $u^{\text{cut-out}} = 25$ ms$^{-1}$, and is plotted in Fig. 1. The parameter $d^{\text{min}}$ is set to $2D$.

The experiments in Sects. 5.2, 5.3, and 5.4 have been carried out on an Intel Core i7-6600U CPU running at 2.80 GHz with four logical processors and 16 GB of RAM. For the experiment in Sect. 5.5, a larger resource is used, an Intel Xeon Gold 6226R CPU running at 2.90 GHz with 32 virtual cores and 640 GB of RAM (DTU Computing Center, 2022).

The selected MILP solver is the commercial branch-and-cut algorithm implemented in IBM ILOG CPLEX Optimization Studio V20.1 (IBM, 2022). Apart from the number of threads and time limit settings, a few other parameters are also set to differently than default values. One is the parameter returning high-quality feasible solutions early in the process, for which, the (CPX_MIPEMPHASIS_HEURISTIC) is activated. The intention is to generate more feasible layouts which is important for the neighborhood search algorithm. Additionally, strong branching is used for variable selection given the large size of the models (CPX_VARSEL_STRONG is selected). The intention is to reduce the size of the search tree and thus the memory requirements compared to default settings.

The number $N$ and positions $\mathbf{p}_i$ for $i \leq N$ of the candidate locations are of course very important parameters for the discrete modelling techniques. A customized automatic strategy based on independently sampling the boundary and interior area of the circular domain $\mathbf{F}$ has been employed. An example of the sampling strategy for these particular case studies giving $N = 467$ is illustrated in Fig. 3.

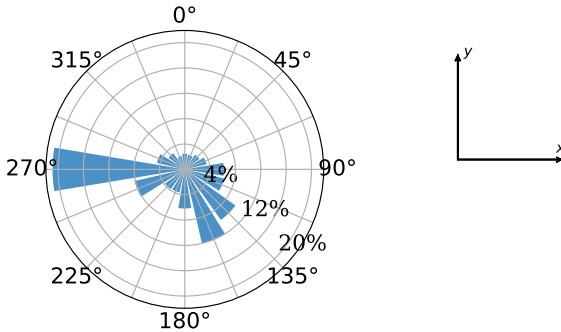

**Figure 2.** Wind rose used in the computational experiments. Taken from open access source IEA Wind Task 37 (2019).

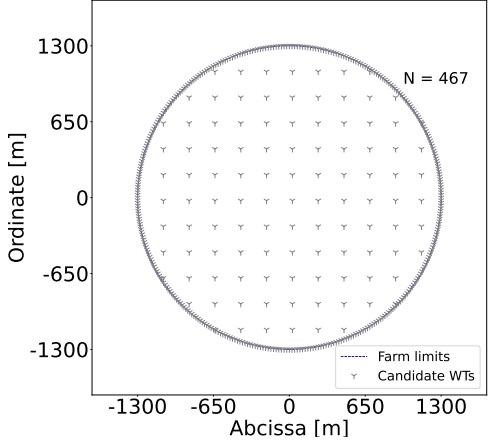

**Figure 3.** Example of generation of WTs candidate locations $N$.

In Fig. 3, the boundary of the circular shape is densely sampled, as a candidate point is defined every natural angle from $0°$ to $359°$, i.e. 360 candidate points are provided since it is intuitively expected that a good portion of the WTs will be placed in the boundaries to decrease wake losses. For the interior, a set of finite parallel line segments are generated and the candidates points are then taken along those segments. In the example of Fig. 3, the slope of the line segments is zero, and the distance between points and lines is equal to $1.7D$.

## 5.1 Correlations

To validate the approach modelled by the MILP formulation of Eq. (23) (i.e. the power-curve-free model), 5000 random feasible WT layouts are created. For each of them, the AEP of Baker et al. (2019), the total theoretical wind speed, $U$, given by Eq. (19), the total wind speed proxy, $\tilde{U}$, defined by Eq. (22), and total wind speed deficit proxy, $\sum_{i=1}^{N} \tau_i$, argument of Eq. (24), are calculated. Although the random way of generating the layouts is biased against high-quality points, the interest is in the general trend in order to assess whether it makes sense to implement the linear proxy objective $\sum_{i=1}^{N} \tau_i$ when optimizing AEP.

In all cases Pearson product-moment linear correlation coefficients from Pearson (1895) are used to extract information from the data and collected in Table 1 for all pairs. This coefficient illustrates the degree to which the movement of pairs of variables is associated in a linear fashion. The correlation plots of Fig. 4 present the graphical representation of the relations for Case I.

The correlation between AEP and the total theoretical wind speed is shown in Fig. 4a for the Case I. The main observation is the very strong linear relation between these two variables as illustrated by the correlation coefficient of $0.97$. Interestingly, this reflects the rather low influence of the WT power curve in obtaining high-quality feasible points. The relation between $U$ and $\tilde{U}$ is represented in Fig. 4b, resulting in an almost identical linear connection between them, as in the previous graph. When one looks into AEP vs $\sum_{i=1}^{N} \tau_i$, however, it is noticeable that the Pearson coefficient decreases to $-0.88$. There is a wider area in the body of points that causes this behaviour. Note that in contrast to the previous two figures, there is a negative correlation because the comparison is done in terms of wind speed deficit instead of total wind speed. In spite of this deterioration, the linear correlation is still considered quite strong. These results motivate the approach where the minimization of a proxy total wind speed deficit can lead to high-quality AEP solutions. The NSH Algorithm 1 helps correcting the imperfect correspondence between these two variables during the optimization routine as reflected in Sect. 4.

**Table 1.** Pearson product-moment linear correlation coefficients for all case studies.

| Case | AEP vs Theoretical wind speed | Theoretical vs Proxy wind speed | AEP vs Proxy wind speed deficit |
|------|:---:|:---:|:---:|
| Case I | 0.97 | 0.96 | -0.88 |
| Case II | 0.97 | 0.95 | -0.85 |
| Case III | 0.96 | 0.88 | -0.72 |

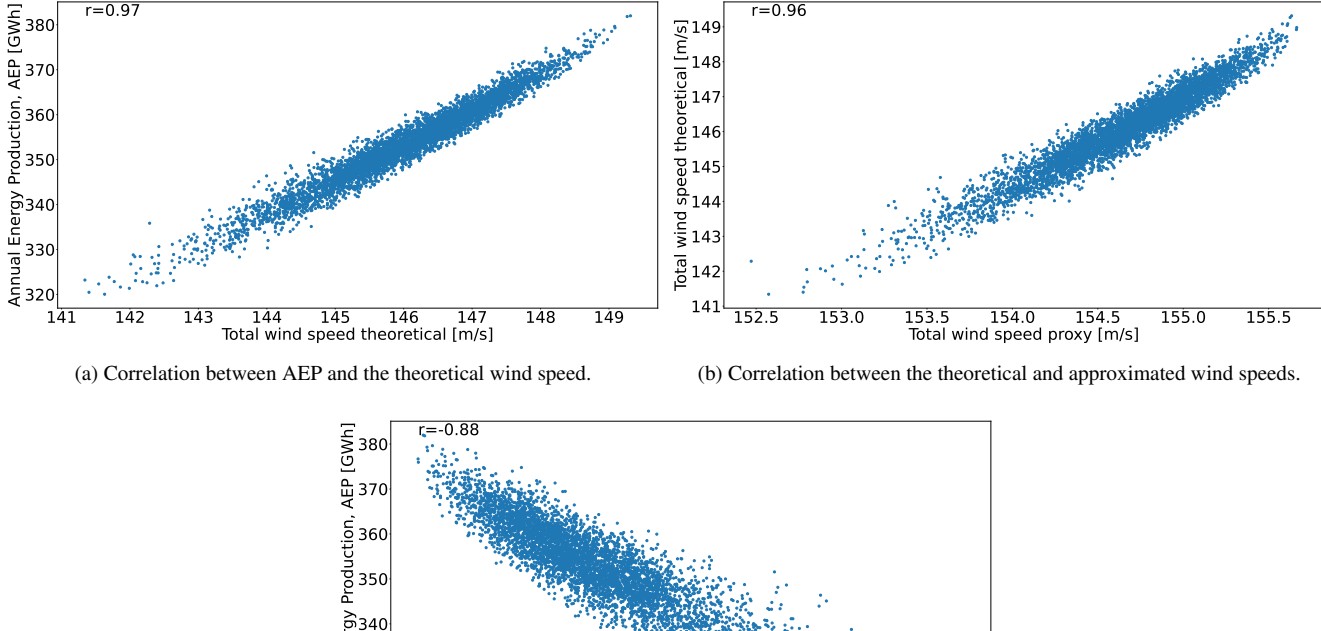

(a) Correlation between AEP and the theoretical wind speed.

(b) Correlation between the theoretical and approximated wind speeds.

(c) Correlation between AEP and the approximated wind speed deficit.

**Figure 4.** Correlation plots for 5000 randomly generated wind turbine layouts for Case I.

The general trends of the correlation plots for Case II are very similar. Correlations between AEP versus theoretical total wind speed (0.97), and theoretical total wind speed versus total wind speed proxy (0.95) are still very strong. Nonetheless, there is a slight decrease between AEP vs total wind speed proxy (down to $-0.85$ from $-0.88$ previously), as the spread for middle velocity values is larger. The linear relation is deemed as satisfactory enough to carry on with the application of model of Eq. (23) with objective function Eq. (24).

The very strong linear relation between AEP and the theoretical total wind speed (0.96) is observed also for Case III, prompting to a very interesting conclusion. Although almost all research in the WFLO space focuses strictly on power modelling (which brings a great deal of complexity due to the non-linear and non-differentiable properties of WT power curve), using an exact model for determining total wind speed as objective function alleviates the computational complexity, while finding high-quality solutions in terms of AEP. However, one should note that deterioration in the correlation still exists, potentially leading to lower quality results.

Likewise, correlations stemming from the proxy to calculate total wind speed deficit are lowered in Case III. This is the case for both with the total wind speed theoretical (0.88) and the AEP ($-0.72$). Keep in mind that the reason to formulate such approximation is to fit in the context of integer programming to leverage theory and state-of-the-art algorithms of this

mature field. However, the price to pay is to lose fidelity to represent the real (true) target to optimize. The deterioration in the correlation of these pairs of variables may also suggest the need to resort to the power-curve-based model for some applications. Whether the price is too high or not is reflected in the reachable solution quality. Sects. 5.2, 5.3, and 5.4 present the optimization results for the cases of fixed number of WTs that will ultimately help to elaborate a final evaluation regarding the adopted modelling technique.

## 5.2 Case I: 16 WTs

This case has a round shape of $1300$ m radius and $n_T = 16$ WTs. The evolution of two of the proposed optimization frameworks is given in Fig. 5 (clock time given in the abscissa). The green line of the full model is obtained after solving the model of Eq. (23) with objective function as in Eq. (24) for $N = 1014$ without implementing the NSH. It represents the incumbent solution in terms of AEP (not total wind speed deficit proxy) obtained by post-processing the CPLEX's log. The blue line results after applying the NSH with the model of Eq. (23) plus objective Eq. (24), and AEP as true objective function in Algorithm 1. The main inputs are $C = \{467, 590, 1014\}$ (set of candidate locations), $T = \{1, 1.5, 2\}$ h (set of max computing times for each candidate location), $V = \{2, 4, 6, 16\}$ (set of neighborhood search sizes). See Sect. 4. These inputs are tuned after evaluating the performance of the method using different values. In general, the first two elements of $C$ consists of moderately big values, relatively close to each other, while the last element is sizeably greater in the search of the best possible solution. Each element $N \in C$ has associated a computing time $T$. Finally, the first elements of $V$ are relatively low values to favour termination of the solver due to optimality, and then they start increasing to refine the search. The red line is for establishing a reference of AEP value, this comes from the best performing method in the survey of IEA Wind Task 37 of Baker et al. (2019), the SNOPT plus Wake Expansion Continuity (WEC) (Thomas and Ning, 2018; Thomas et al., 2022b). Time evolution for the SNOPT+WEC is not reflected in this graph, as this information is unavailable. Results for the benchmark against a testbed of different algorithms are available in Table 2.

The NSH computing time results in Fig. 5 do not reflect the instant where the incumbent is found, but the time progress of this algorithm, which is dependent on the execution of the MILP solver at each iteration. Table A1 contains information about the values of $N$, $K$, $T$, and termination criterion of the solver after each iteration $\kappa$ of the NSH Algorithm 1 (beginning from point 2 where $\kappa = 1$). This means that, in iterations where the termination criterion is time (and not optimality), one could fine-tune $T$ for an earlier stop, shortening the total time. This is particularly more relevant in cases where internal heuristics of the solver are activated at the root node of the search tree, coming up with the largest portion of solutions very early in the process. Consequently, the total computing time, for all cases, is conservative and should be taken as an approximated reference.

The initial layout (point 1), labelled in Fig. 6a, is set up by arbitrarily by picking up candidate locations around the circular boundary; this layout has an AEP of $387$ GWh. From now on, the presented percentages are calculated with respect to the last commented AEP improvement. Between points 2 to 7 where $N = 467$ and $K = 2$, the models are solved to optimality (gap of $0\%$), and the solution is improved by $2.92\%$ in only $56$ s. After a short plateau, the solution is markedly refined by $2.96\%$ from point 10 to 13 by performing a search of the domain with $K = 16$, and restarting the model every $1$ h with a new warm-starting.

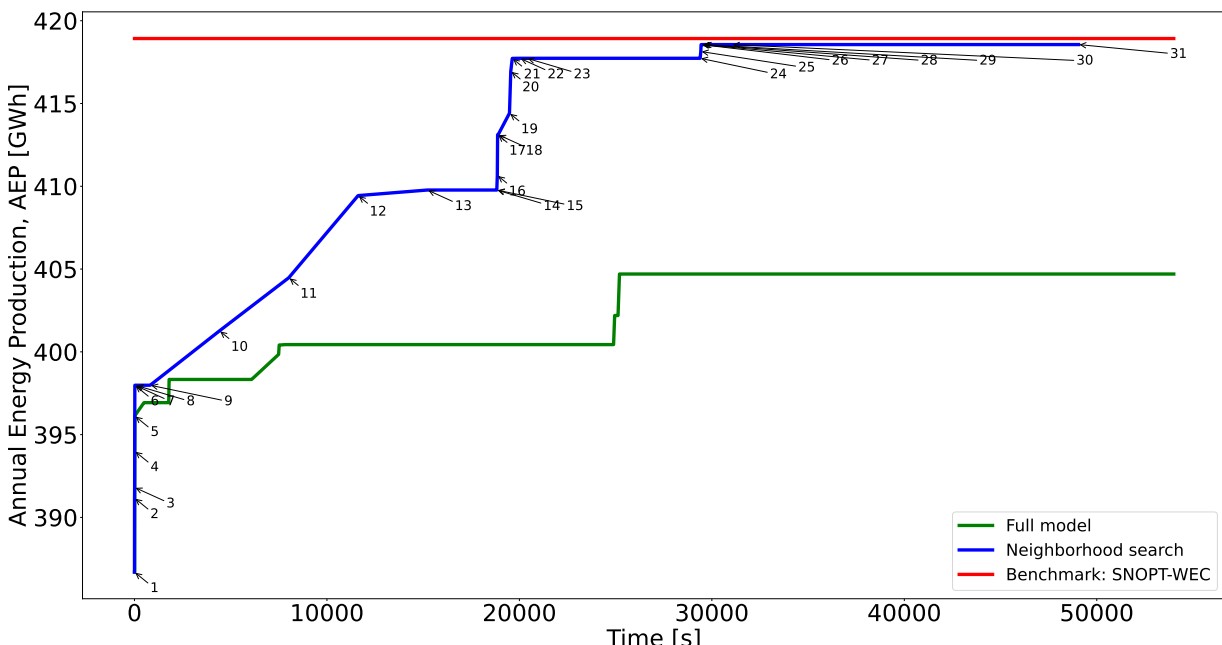

**Figure 5.** Performance of two different optimization approaches for Case I and comparison with existing best benchmark results. See Table A1 for detailed information about each numbered point in the blue curve.

The next considerable jump happens for $N = 590$ and $2 \leq K \leq 6$ in around $20 \min$, elevating the AEP by $1.94\%$. After, again, a plateau without improvements, when $N$ reaches its maximum value of $1014$, the solution is maximized to the final value of $418 \, \mathrm{GWh}$ during the lowest values of $K$. For this particular instance, the greatest value of $K = 16$ is exploited for the lowest number of candidate points $N$, where the largest improvement comes up.

The benefit of the proposed neighborhood search strategy is shown in Fig. 5. Solving the full model is significantly slower, leading actually to a worse solution ($3.31\%$ lower). The capacity of the NSH to iterate over different values of candidate points $N$ and search sizes $K$ brings alone not only improvements in terms of solution time and solution quality, but also less computational resources as the RAM memory generally scales faster when solving the single model.

The initial and final solution layouts for this case study are illustrated in Fig. 6. The importance of finely sampling the boundaries of the available area is evident in Fig. 6b, because 7 out of the 16 WTs are placed in that subdomain.

Finally, Table 2 compares the proposed method to a large number of different approaches from the IEA37 reference study (Baker et al., 2019). The results for all case studies are presented, where I, II, and III make reference to cases from this section, Sect. 5.3, and Sect. 5.4, respectively.

The third column of Table 2 reports the difference of the AEP with respect to the proposed method for the smallest case study. The resulting AEP is better than almost all the other alternatives, except to the SNOPT+WEC, where a nearly identical objective value is achieved. When directly comparing to to typical metaheuristics (genetic algorithm, particles swarm optimization, etc), that do not use explicit gradients information, the presented method seems to perform well, being able of determining a similar

layout quality in less than 2 h, which is usually a competitive time compared to these kinds of population-based algorithms. In a broader context, beyond the presented numerical comparisons, discrete optimization approaches, as the MILP ones presented in this manuscript, could be formulated to cope with problem definitions with required functionalities that in theory continuous optimization methods can not support (or at least the implementation becomes strenuous).

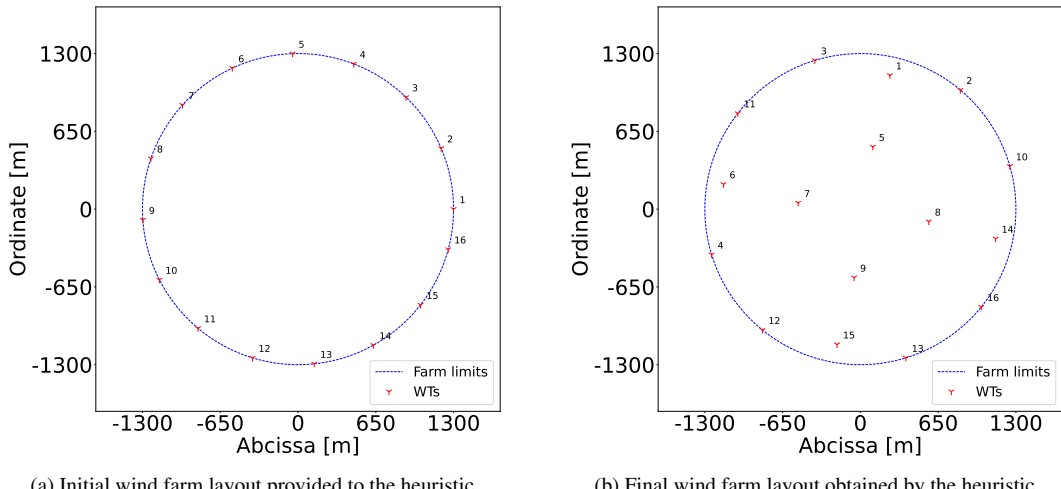

(a) Initial wind farm layout provided to the heuristic.  (b) Final wind farm layout obtained by the heuristic.

**Figure 6.** Generated wind farm layouts for the benchmark Case I with 16 turbines.

**Table 2.** Results for all three benchmark cases from other algorithms (G, gradient-based and GF, gradient-free) obtained while allowing WT locations to vary continuously. Values reproduced from Baker et al. (2019). The difference column shows how the proposed heuristic with the power-curve-free model performs in comparison. Negative percentages means that the proposed method performs better than the corresponding algorithm.

| Method | AEP I [GWh] | Diff. I [%] | AEP II [GWh] | Diff. II [%] | AEP III [GWh] | Diff. III [%] |
|---|---|---|---|---|---|---|
| SNOPT+WEC (G) | 418.92 | 0.09 | 863.68 | -0.19 | 1513.31 | 0.85 |
| fmincon (G) | 414.14 | -1.06 | 820.39 | -5.19 | 1336.16 | -10.95 |
| SNOPT (G) | 412.25 | -1.51 | 846.36 | -2.19 | 1476.69 | -1.59 |
| SNOPT (G) | 411.18 | -1.76 | 844.28 | -2.43 | 1445.97 | -3.64 |
| Preconditioned SQP (G) | 409.69 | -2.12 | 849.37 | -1.84 | 1506.39 | 0.39 |
| Mul.interior-point (G) | 408.36 | -2.44 | 851.63 | -1.58 | 1480.85 | -1.31 |
| Full pseudo-gradient (GF) | 402.32 | -3.88 | 828.75 | -4.23 | 1455.08 | -3.03 |
| Basic genetic algorithm (GF) | 392.59 | -6.20 | 777.48 | -10.15 | 1332.88 | -11.17 |
| Simple particle swarm (GF) | 388.76 | -7.12 | 776 | -10.32 | 1364.94 | -9.04 |
| Simple pseudo-gradient (GF) | 388.34 | -7.22 | 813.54 | -5.98 | 1422.27 | -5.22 |

The power-curve-based model of Eq. (18) within the NSH using the same AEP formulation as true objective function, provides a solution $1.18\%$ lower in objective value in around $36$ h using the computer system with 32 virtual cores. Although the quality of the layout is very close to the one schematized in Fig. 6b, the larger computational resources favour implementing the power-curve-free model for problems with fixed number of WTs. Therefore, Sects. 5.3 and 5.4 present only the results reached after the application of the power-curve-free model embedded into the NSH.

## 5.3    Case II: 36 WTs

This case has a round shape of $2000$ m radius and $n_T = 36$ WTs. The evolution of the proposed methods, and the initial and final WT layouts are plotted in Fig. 7 and Fig. 8, respectively. Table B1 displays the data linked to each point of Fig. 7. Main inputs are $\boldsymbol{C} = \{477, 684, 1907\}$, $\boldsymbol{T} = \{1, 1.5, 2\}$ h, $\boldsymbol{V} = \{2, 4, 8, 16, 36\}$. The blue line (model of Eq. (23) with objective function Eq. (24) plus NSH Algorithm 1) has clearly three sectors stemming from each value of $N \in \boldsymbol{C}$. The initial WT layout (Fig. 8a)
- also determined by choosing roughly equidistant candidate locations in the boundary - has an AEP of $796$ GWh. As for Case I, improvement percentages are calculated using the last commented step as the baseline. After seven NSH iterations (point 8) in $41$ s, the incumbent is improved by $1.84\%$, when $N = 477$ and $2 \leq K \leq 4$, being able to solve each model instantation to optimality.

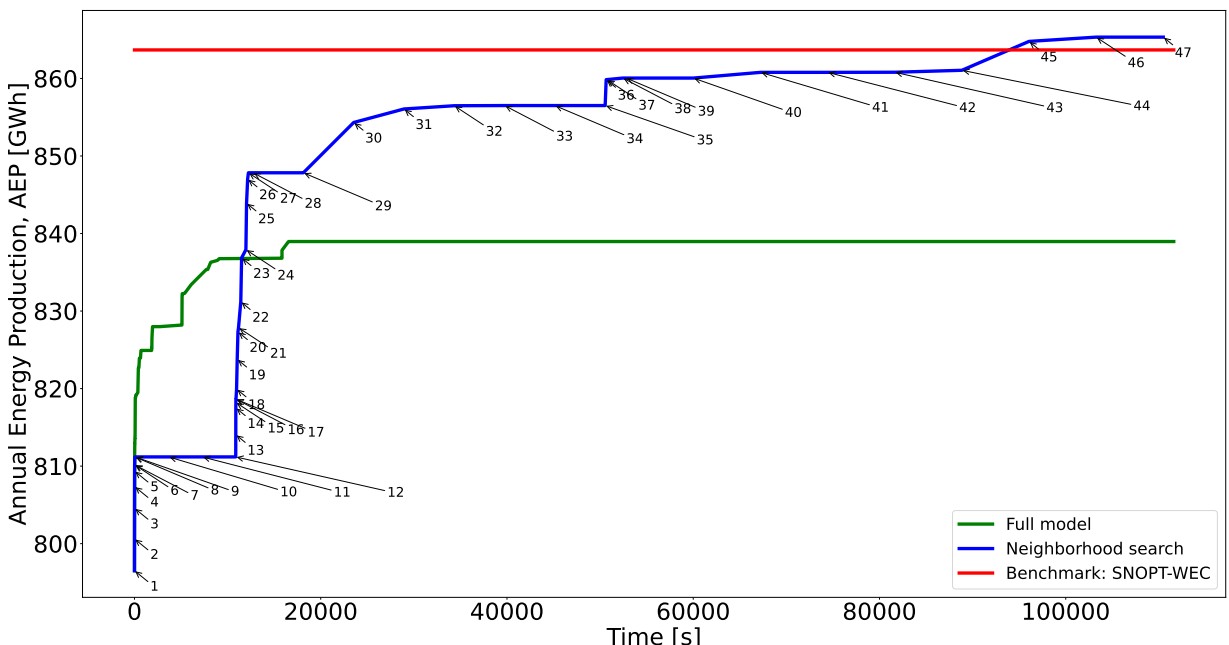

**Figure 7.** Performance of two different optimization approaches for Case II and comparison with existing best benchmark results. See Table B1 for detailed information about each numbered point in the blue curve.

After a three-hours-plateau linked to $8 \leq K \leq 36$ (four iterations), $N$ is raised to $684$, resulting in the largest AEP en-
hancement, as shown in Fig. 7. The energy production increases with $4.51\%$ after only $23$ min in point 27. This noticeable

improvement comes after solving to optimality models with rather small neighborhood search sizes $2 \le K \le 4$. The convenience of allowing large neighborhood search sizes as $K = 16$ or $K = 36$ is reflected from this moment. From point 30 to 33 (6 h) with $K = 16$ the incumbent is slowly boosted by nearly $1\%$. Again, after a three-hours-plateau, $N$ becomes equal to 1907, and in around 32 min for $2 \le K \le 4$, the AEP is augmented by $0.41\%$. Then, the large neighborhood search starts for $K = 16$ and $K = 36$, and after a total of 16 h, the final solution of 865 GWh (increment of $0.61\%$) is achieved (Fig. 8b).

The full model (i.e. without implementing the NSH algorithm) initially provides better solutions within the first 3 h, but then lags behind in solution quality compared to the NSH algorithm in the long run (lower $3.05\%$), as shown in in Fig. 7.

For this case, the proposed method reaches the best solution, as shown in the fifth column of Table 2. The SNOPT+WEC is again the closest contender. When uniquely comparing to GF methods, the proposed method matches the best solution from those algorithms in around 3 h, which is generally a reasonable computing time compared to methods where gradients are not explicitly utilized in the optimization process, especially to metaheuristics as genetic algorithm or swarm optimization.

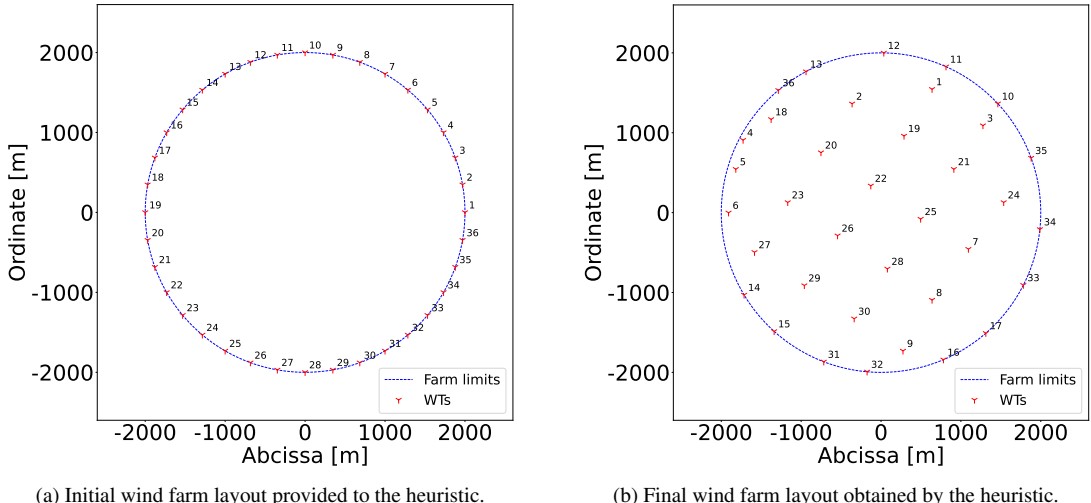

(a) Initial wind farm layout provided to the heuristic.    (b) Final wind farm layout obtained by the heuristic.

**Figure 8.** Generated wind farm layouts for the benchmark Case II with 36 wind turbines.

## 5.4 Case III: 64 WTs

This case has a round shape of 3000 m radius and $n_T = 64$ WTs. The evolution of the proposed methods, and the initial and final WT layouts are displayed in Fig. 9 and Fig. 10, respectively. Table C1 displays the data linked to each point of Fig. 9. Main inputs are $C = \{625, 1017, 2741\}$, $T = \{1, 1.5, 2\}$ h, $V = \{2, 4, 8, 16, 32, 64\}$. Note that in comparison the number of elements of $V$ has been increased by one after each study case. This has been done taking into account the number of WTs. Likewise, the values of $N \in C$ are larger to cover for the wider project areas.

Comparing blue lines of Fig. 5, Fig. 7, and Fig. 9 is evident that for the last case the curve shows less sudden increases. The largest change occurs after 27 s where the initial solution (Fig. 10a) with AEP of 1395 GWh is improved by $3.18\%$ for

$N = 625$ and $K = 2$ up to point 9, reaching optimality in few seconds. With $4 \leq K \leq 8$ the model instantiations are solved to optimality in minutes, obtaining a solution improved by $0.18\%$.

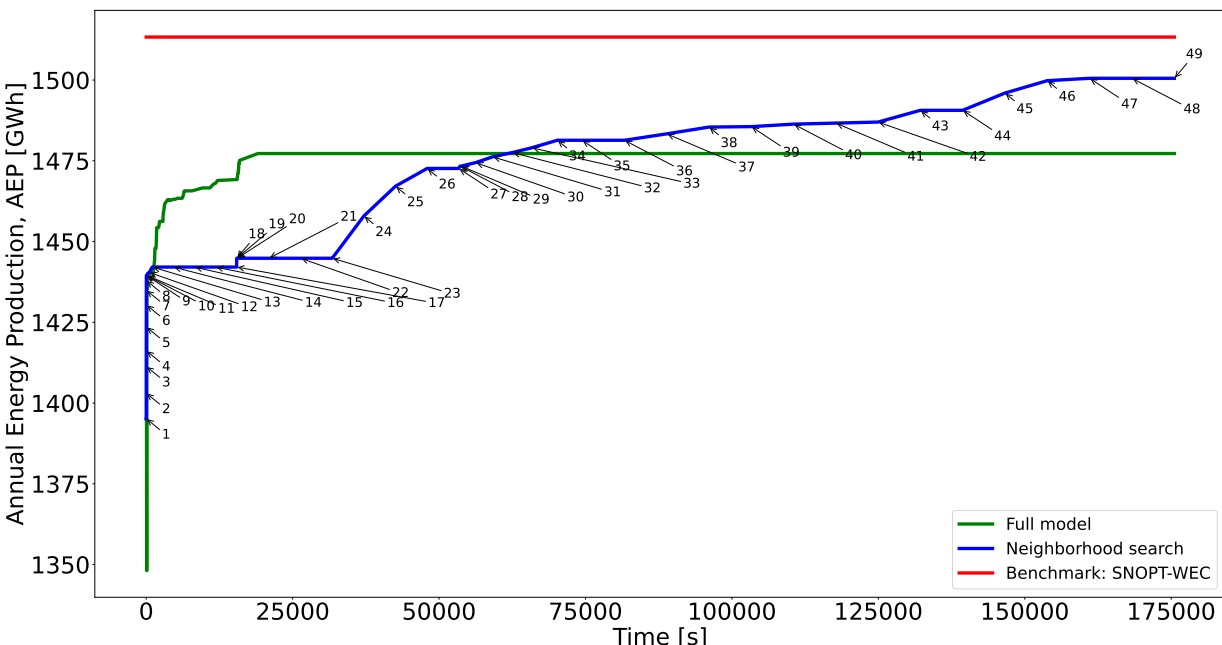

**Figure 9.** Performance of two different optimization approaches for Case III and comparison with existing best benchmark results. See Table C1 for detailed information about each numbered point in the blue curve.

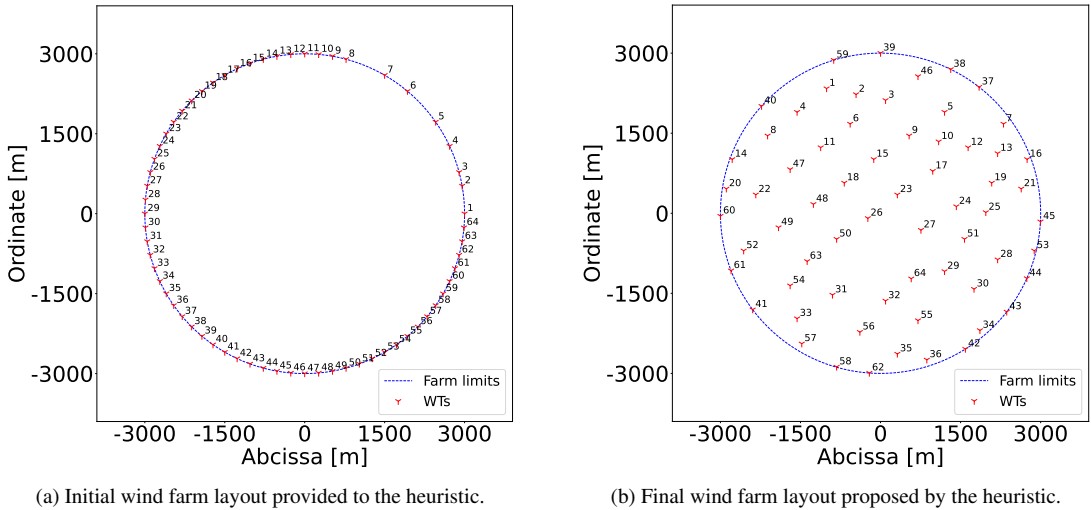

(a) Initial wind farm layout provided to the heuristic.      (b) Final wind farm layout proposed by the heuristic.

**Figure 10.** Generated wind farm layouts for benchmark Case III with 64 wind turbines.

After point 13 in Fig. 9, one notes a plateau without improvement for $N = 625$ and $K \geq 16$, i.e. a large neighborhood search does not lead to further enhancements. Due to this, $N$ is enlarged to 1017, where the second largest boost (increase of 2.12%) comes, with the largest search size ($K = 64$) resulting in the best improvement. This enhancement occurs after 13 h of starting the NSH (point 26). From point 28, $N = 2741$ and for $2 \leq K \leq 4$ the solver reaches optimality; slowly converging to the final solution of 1500 GWh (Fig. 10b).

Seventh column of Table 2 shows that the SNOPT+WEC and the preconditioned SQP provide slightly better layouts than the proposed method. However, the algorithm provides feasible layouts that improve the objective compared to all the gradient-free approaches.

## 5.5  Case IV: 10-50 WTs

Although in most projects today the total capacity for grid connection is decided already in the early planning phases, in the future one can envisage situations where flexibility in optimizing the number of wind turbines in a project would yield benefits.

Even if the power-curve-free model (Sect. 3.2) exhibits a quite good performance in terms of AEP and computing time for fixed number of WTs (when AEP and NPV are basically the same metric), it is not very well suited when variable number of wind turbines are considered. Based on computational experiments not included in the paper, the power-curve-free model embedded in the NSH terminates too early in the search process, resulting in a worse solution than the alternative discussed below.

For such an optimization, the power-curve-based mathematical program of Sect. 3.1 may be handy as the number of generators is allowed to vary between a lower and upper bound, $n^{\min}$ and $n^{\max}$, respectively. For illustration, a domain defined by a circle with radius 1300 m, and variable number of WTs between 10 and 50 are utilized. These parameters are set relatively arbitrarily but with sufficient distance to reasonably expect that the limits are not reached. The aim is to illustrate the ability of the method in reaching non-trivial solutions, resulting in a an optimized design with an intermediate number of wind turbines.

Keep in mind that for this case, a linear superposition model for the AEP component in the NPV calculation is considered. In this sense, the original WT power curve as depicted in Fig. 1 is used. NPV is the true objective function when applying the NSH Algorithm 1. The modified objective function of MILP model of Eq. (18) for this case has the form (Cogency, 2014):

$$\underset{\xi,\eta,u}{\text{maximize}} \; -\sum_{i=1}^{N} c_{\mathrm{wt}}\xi_i + 8760 \sum_{y=1}^{Y} \sum_{i=1}^{N} \sum_{j,k} \sum_{l=1}^{m+2} \frac{c_{\mathrm{e}} w_{jk} \eta_{ijk}^{l} p(u_{\mathrm{m}}^{l})}{(1+r)^y} \tag{25}$$

where $c_{\mathrm{wt}}$ is the cost per WT in mEUR, $c_{\mathrm{e}}$ the energy price in $\mathrm{mEURMWh^{-1}}$, $r$ is the discount rate in %, and $Y$ is the number of years of lifetime of the project. For this case study, values of $c_{\mathrm{wt}} = 6.7$ mEUR (Mishnaevsky Jr and Thomsen, 2020), $c_{\mathrm{e}} = 0.00015 \; \mathrm{mEURMWh^{-1}}$ (Nord Pool, 2022), $r = 5\%$, and $Y = 20$ are assumed. The general form of the NPV equation as per Cogency (2014) is defined by the sum of the present value of cash flows (Discounted Cash Flow, DCF) of a project under analysis. In Eq. (25), the first sum is a negative cash flow representing purchase of the WTs at the construction stage of the project, while the next term represents positive cash flows coming from trading the electricity in the market. Because of the

additive nature of the NPV metric and since the focus is on evaluating investment vs revenues, by maximizing Eq. (25), a fully comprehensive NPV metric is equivalently improved.

The model of Eq. (18) with modified objective function Eq. (25), embedded in the NSH Algorithm 1 with NPV as the target function is executed in three runs. For the first run the number of turbines is fixed to $n^{\min} = n^{\max} = 10$, while for the second the number of turbines remains fixed but is increased to $n^{\min} = n^{\max} = 50$. For the third run the number of wind turbines is allowed to vary between $n^{\min} = 10$ and $n^{\max} = 50$. The Algorithm 1 input parameters are $C = \{467, 590, 1014\}$, $T = \{1, 1.5, 2\}$ h, $V = \{2, 4, 6, 8, 24\}$. The results are plotted in Figure 11.

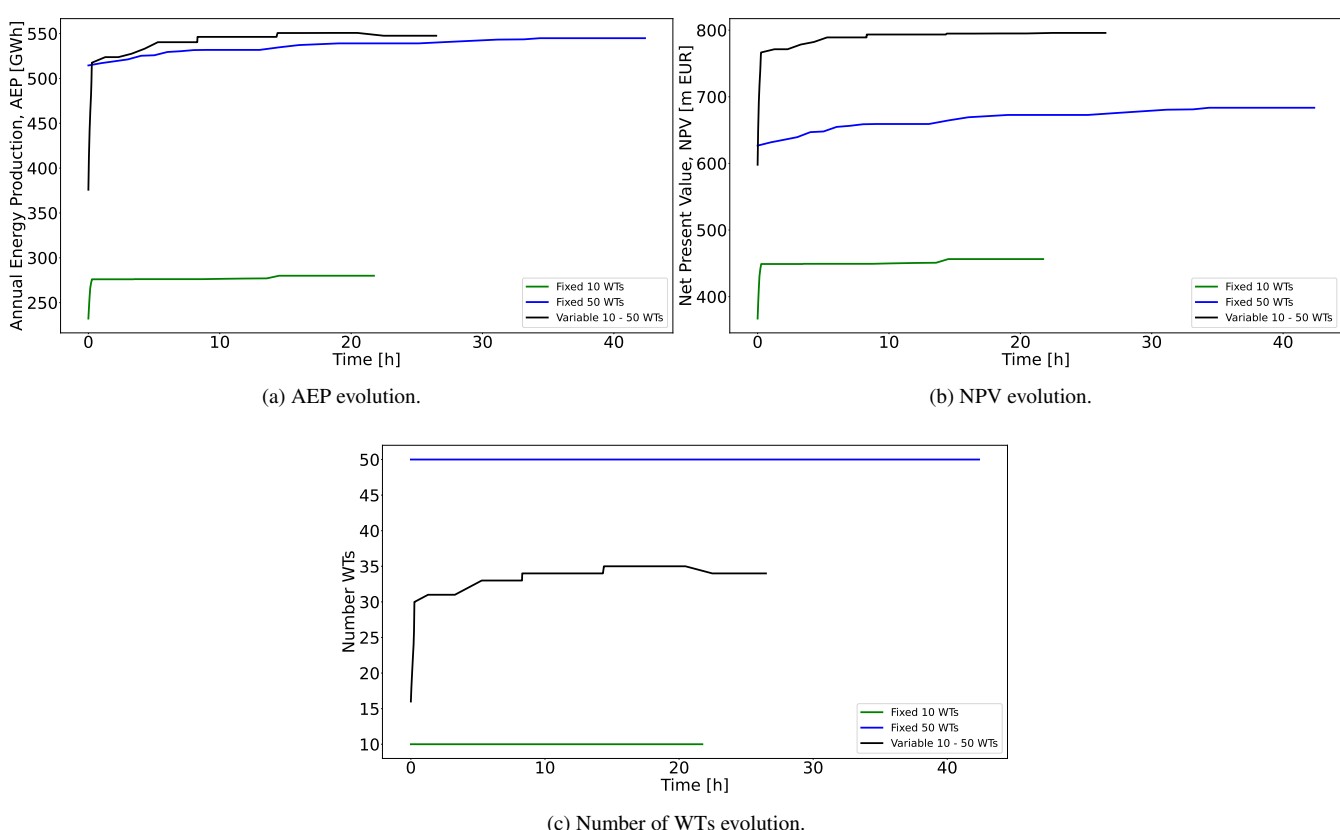

(a) AEP evolution.

(b) NPV evolution.

(c) Number of WTs evolution.

**Figure 11.** Evolution of the AEP, NPV, and number of WTs for the three simulations. The green lines are results for the optimization program with fixed number of WTs equal to 10, the blue ones equal to 50, and the black ones for the optimization program with variable number of WTs between 10 and 50.

When the number of turbines is fixed to 10, the NPV evolution (green line in Fig. 11b) is driven by the AEP (green line in Fig. 11a). Both curves are monotonically increasing, reaching a final value of NPV of $= 456.40$ mEUR. The same behaviour is visible for $n_{\mathrm{T}} = 50$, although the final NPV is greater (683.53 mEUR), see blue line Figure 11b. In the second study, the positive difference in DCF from the revenues surpasses the associated extra investment costs from the additional 40 wind

turbines considered. The significant increase in the number of WTs doubles the computing time, due to the large increase in the number of variables, selecting 50 WTs entails significantly more possible combinations of valid points.

An interesting question is whether there is a larger NPV in between the bounds of WT number. For the optimization program with variable number of WTs, the evolution of the WTs number in Fig. 11c and the AEP in Fig. 11a (see black lines in these figures) exhibits a perfect correspondence. The more WTs the larger AEP, in spite of the increased wake losses. The curves increase in time, up to a point where the model estimates that further increase of WTs would not lead to a better NPV. The final number of WTs is 34. The NPV evolution in Fig. 11b (black line) naturally only improves with time, resulting in a final value of 795.86 mEUR. Note that the NPV in this case is greater than when a larger number of WTs (i.e. 50) was considered and of course when only 10 were considered. Interestingly, the optimization program with 50 fixed number of WTs finds a final solution with AEP very close to that from the variable number program, being the solution of the former 0.50% lower than the latter, but requiring more WTs, and hence more investment (47% more). The final NPV value of the variable number model is 16.43% greater than the one with fixed 50 WTs. These figures could be expected to be similar even in situations where lower AEPs are obtained, if that compensates by augmenting overall financial metrics as the NPV.

This result shows the benefit of having optimization models that support variable number of WTs and accounting for metrics beyond AEP. The advantages may become even more pronounced for more complex situations, as for instance, if the WT investment costs are dependent on the exact installation area or different WT sizes are considered.

## 6 Discussions

The two models proposed in this manuscript have many of the characteristics of mixed integer linear programming models. They require significant computational time and memory and exhibit rather low tractability and scalability for global optimization algorithms.

The power-curve-based model, albeit requiring large computational resources, manages to provide reasonably good solutions for small-sized problem, being only 1.18% lower than its power-curve-free counterpart for the 16 WTs case and 4.41% for the 36 WTs case. This diminishing efficiency is to be expected, given the large number of variables and constraints. The power-curve-free model on the other hand, along with the heuristic, is much faster due to its more compact formulation. This translates into the ability to be highly competitive compared to a large set of benchmark algorithms. In situations where there is an interest for optimizing metrics beyond AEP, such as the NPV, the power-curve-based model becomes very useful given its intrinsic capacity to support this kind of objective functions.

It should be mentioned that there are limitations in the wake models used compared to recent ones (Thomas et al., 2022b). For example, the wake model used in this manuscript does not consider the changes in the turbulence intensity or thrust coefficient variations from wind speed variations inside the wind farm. It is uncertain if using wake models like the ones in Thomas et al. (2022b) would still allow an integer linear programming formulation or approximation of the WFLO problem. It is also uncertain the impact on the final solution quality these detailed modelling aspects imply. These questions are left for future work.

Notwithstanding the listed shortcomings, it is enthralling that these models, in combination with the neighborhood search heuristic, are able to match and in some cases improve the results obtained when considering the turbine positions as continuous variables (see Table 2). This opens the door to experimenting case studies with functionalities easily adaptable to discrete parametrization techniques, which can be very challenging for continuous variable modelling approaches.

## 7  Conclusions

This manuscript contributes both methodologically and empirically to address the WFLO problem. A neighborhood search heuristic embedding integer programming formulations is proposed. For both presented formulations presented in the manuscript, the step-wise power curve and power-curve-free, the heuristic notably improves a single execution of full models when calling a state-of-the-art branch-and-cut solver in terms of solution quality. An improvement of up to $3.42\%$ in the AEP is achieved by applying the neighborhood search strategy for cases where the WTs number is fixed compared to solving the full model.

Another important takeaway is the satisfactory performance of the power-curve-free model, which uses an approximation of the total wind speed deficit, when (implicitly) optimizing for AEP. This is due to the good correlation between the two measures, and the correction capability of the heuristic. For the classic WFLO problem definition, the proposed model is able to considerably improve (from $1\%$ to around $10\%$) the AEP compared to benchmark results by multiple gradient-based and gradient-free algorithms. Even when directly compared to methods implementing a continuous variables technique, the proposed heuristic provides similar or even better results. These are very promising results that would enable to get high-quality solutions for problem instances where continuous variables modelling approaches may not be able to run or provide with good incumbents.

Finally, the model with explicit representation of the power curve embedded within the neighborhood search heuristic is able to propose non-trivial solutions when implementing objective functions beyond AEP, such as NPV. For these cases, the trade-off between energy revenues and investment costs is studied. For example, the model suggests that is installing a lower number of wind turbines than the allowed would results in a better NPV value, with comparable AEP.

# Appendix A: Case I

**Table A1.** Information about the values of $N, K, T$, and termination criterion of the solver after each point in Fig. 5.

| Point | $K, N$ | | Termination criterion | Point | $K, N$ | | Termination criterion |
|---|---|---|---|---|---|---|---|
| 1 | Initial point | | - | 17 | $K = 4$ | $N = 590$ | opt. [0.24 min] |
| 2 | $K = 2$ | $N = 467$ | opt. [0.05 min] | 18 | $K = 4$ | $N = 590$ | opt. [0.46 min] |
| 3 | $K = 2$ | $N = 467$ | opt. [0.06 min] | 19 | $K = 6$ | $N = 590$ | opt. [9.79 min] |
| 4 | $K = 2$ | $N = 467$ | opt. [0.04 min] | 20 | $K = 6$ | $N = 590$ | opt. [1.10 min] |
| 5 | $K = 2$ | $N = 467$ | opt. [0.06 min] | 21 | $K = 6$ | $N = 590$ | opt. [1.29 min] |
| 6 | $K = 2$ | $N = 467$ | opt. [0.05 min] | 22 | $K = 6$ | $N = 590$ | opt. [6.64 min] |
| 7 | $K = 2$ | $N = 467$ | opt. [0.07 min] | 23 | $K = 6$ | $N = 590$ | opt. [6 min] |
| 8 | $K = 4$ | $N = 467$ | opt. [0.49 min] | 24 | $K = 16$ | $N = 590$ | 1.5 h |
| 9 | $K = 6$ | $N = 467$ | opt. [12.69 min] | 25 | $K = 2$ | $N = 1014$ | opt. [0.53 min] |
| 10 | $K = 16$ | $N = 467$ | 1 h | 26 | $K = 2$ | $N = 1014$ | opt. [0.20 min] |
| 11 | $K = 16$ | $N = 467$ | 1 h | 27 | $K = 2$ | $N = 1014$ | opt. [0.20 min] |
| 12 | $K = 16$ | $N = 467$ | 1 h | 28 | $K = 4$ | $N = 1014$ | opt. [0.95 min] |
| 13 | $K = 16$ | $N = 467$ | 1 h | 29 | $K = 4$ | $N = 1014$ | opt. [1.10 min] |
| 14 | $K = 16$ | $N = 467$ | 1 h | 30 | $K = 6$ | $N = 1014$ | opt. [24.37 min] |
| 15 | $K = 2$ | $N = 590$ | opt. [0.06 min] | 31 | $K = 16$ | $N = 1014$ | 2 h |
| 16 | $K = 4$ | $N = 590$ | opt. [0.43 min] | | | | |

 **Appendix B: Case II**

**Table B1.** Information about the values of $N$, $K$, $T$, and termination criterion of the solver after each point in Fig. 7.

| Point | $K$, $N$ | | Termination criterion | Point | $K$, $N$ | | Termination criterion |
|---|---|---|---|---|---|---|---|
| 1 | Initial point | | - | 25 | $K = 4$ | $N = 684$ | opt. [1.15 min] |
| 2 | $K = 2$ | $N = 477$ | opt. [0.03 min] | 26 | $K = 4$ | $N = 684$ | opt. [1.90 min] |
| 3 | $K = 2$ | $N = 477$ | opt. [0.03 min] | 27 | $K = 4$ | $N = 684$ | opt. [1.38 min] |
| 4 | $K = 2$ | $N = 477$ | opt. [0.04 min] | 28 | $K = 4$ | $N = 684$ | opt. [8.31 min] |
| 5 | $K = 2$ | $N = 477$ | opt. [0.05 min] | 29 | $K = 8$ | $N = 684$ | 1.5 h |
| 6 | $K = 2$ | $N = 477$ | opt. [0.05 min] | 30 | $K = 16$ | $N = 684$ | 1.5 h |
| 7 | $K = 2$ | $N = 477$ | opt. [0.04 min] | 31 | $K = 16$ | $N = 684$ | 1.5 h |
| 8 | $K = 4$ | $N = 477$ | opt. [0.37 min] | 32 | $K = 16$ | $N = 684$ | 1.5 h |
| 9 | $K = 4$ | $N = 477$ | opt. [0.49 min] | 33 | $K = 16$ | $N = 684$ | 1.5 h |
| 10 | $K = 8$ | $N = 477$ | 1 h | 34 | $K = 16$ | $N = 684$ | 1.5 h |
| 11 | $K = 16$ | $N = 477$ | 1 h | 35 | $K = 36$ | $N = 684$ | 1.5 h |
| 12 | $K = 36$ | $N = 477$ | 1 h | 36 | $K = 2$ | N=1907 | opt. [1.65 min] |
| 13 | $K = 2$ | $N = 684$ | opt. [0.07 min] | 37 | $K = 2$ | N=1907 | opt. [1.12 min] |
| 14 | $K = 2$ | $N = 684$ | opt. [0.07 min] | 38 | $K = 4$ | N=1907 | opt. [28.40 min] |
| 15 | $K = 2$ | $N = 684$ | opt. [0.07 min] | 39 | $K = 4$ | N=1907 | opt. [5.97 min] |
| 16 | $K = 2$ | $N = 684$ | opt. [0.08 min] | 40 | $K = 8$ | N=1907 | 2 h |
| 17 | $K = 2$ | $N = 684$ | opt. [0.09 min] | 41 | $K = 16$ | N=1907 | 2 h |
| 18 | $K = 4$ | $N = 684$ | opt. [1 min] | 42 | $K = 16$ | N=1907 | 2 h |
| 19 | $K = 4$ | $N = 684$ | opt. [1 min] | 43 | $K = 36$ | N=1907 | 2 h |
| 20 | $K = 4$ | $N = 684$ | opt. [1.33 min] | 44 | $K = 36$ | N=1907 | 2 h |
| 21 | $K = 4$ | $N = 684$ | opt. [0.98 min] | 45 | $K = 36$ | N=1907 | 2 h |
| 22 | $K = 4$ | $N = 684$ | opt. [4.05 min] | 46 | $K = 36$ | N=1907 | 2 h |
| 23 | $K = 4$ | $N = 684$ | opt. [1.47 min] | 47 | $K = 36$ | N=1907 | 2 h |
| 24 | $K = 4$ | $N = 684$ | opt. [7.65 min] | | | | |

## Appendix C: Case III

**Table C1.** Information about the values of $N$, $K$, $T$, and termination criterion of the solver after each point in Fig. 9.

| Point | $K$, $N$ | | Termination criterion | Point | $K$, $N$ | | Termination criterion |
|---|---|---|---|---|---|---|---|
| 1 | Initial point | | | 26 | $K = 64$ | $N = 1017$ | 1.5 h |
| 2 | $K = 2$ | $N = 625$ | opt. [0.03 min] | 27 | $K = 64$ | $N = 1017$ | 1.5 h |
| 3 | $K = 2$ | $N = 625$ | opt. [0.03 min] | 28 | $K = 2$ | $N = 2741$ | opt. [3.22 min] |
| 4 | $K = 2$ | $N = 625$ | opt. [0.03 min] | 29 | $K = 2$ | $N = 2741$ | opt. [2.93 min] |
| 5 | $K = 2$ | $N = 625$ | opt. [0.04 min] | 30 | $K = 4$ | $N = 2741$ | opt. [40.82 min] |
| 6 | $K = 2$ | $N = 625$ | opt. [0.05 min] | 31 | $K = 4$ | $N = 2741$ | opt. [47.99 min] |
| 7 | $K = 2$ | $N = 625$ | opt. [0.04 min] | 32 | $K = 4$ | $N = 2741$ | opt. [55.95 min] |
| 8 | $K = 2$ | $N = 625$ | opt. [0.04 min] | 33 | $K = 4$ | $N = 2741$ | opt. [54.74 min] |
| 9 | $K = 2$ | $N = 625$ | opt. [0.05 min] | 34 | $K = 4$ | $N = 2741$ | opt. [72.46 min] |
| 10 | $K = 2$ | $N = 625$ | opt. [0.04 min] | 35 | $K = 4$ | $N = 2741$ | opt. [69.85 min] |
| 11 | $K = 4$ | $N = 625$ | opt. [0.33 min] | 36 | $K = 8$ | $N = 2741$ | 2 h |
| 12 | $K = 8$ | $N = 625$ | opt. [7.92 min] | 37 | $K = 16$ | $N = 2741$ | 2 h |
| 13 | $K = 8$ | $N = 625$ | opt. [8.31 min] | 38 | $K = 16$ | $N = 2741$ | 2 h |
| 14 | $K = 8$ | $N = 625$ | 1 h | 39 | $K = 16$ | $N = 2741$ | 2 h |
| 15 | $K = 16$ | $N = 625$ | 1 h | 40 | $K = 16$ | $N = 2741$ | 2 h |
| 16 | $K = 32$ | $N = 625$ | 1 h | 41 | $K = 16$ | $N = 2741$ | 2 h |
| 17 | $K = 64$ | $N = 625$ | 1 h | 42 | $K = 16$ | $N = 2741$ | 2 h |
| 18 | $K = 2$ | $N = 1017$ | opt. [0.17 min] | 43 | $K = 16$ | $N = 2741$ | 2 h |
| 19 | $K = 2$ | $N = 1017$ | opt. [0.26 min] | 44 | $K = 16$ | $N = 2741$ | 2 h |
| 20 | $K = 4$ | $N = 1017$ | opt. [1.35 min] | 45 | $K = 32$ | $N = 2741$ | 2 h |
| 21 | $K = 8$ | $N = 1017$ | 1.5 h | 46 | $K = 32$ | $N = 2741$ | 2 h |
| 22 | $K = 16$ | $N = 1017$ | 1.5 h | 47 | $K = 32$ | $N = 2741$ | 2 h |
| 23 | $K = 32$ | $N = 1017$ | 1.5 h | 48 | $K = 32$ | $N = 2741$ | 2 h |
| 24 | $K = 64$ | $N = 1017$ | 1.5 h | 49 | $K = 64$ | $N = 2741$ | 2 h |
| 25 | $K = 64$ | $N = 1017$ | 1.5 h | | | | |

*Code availability.* Available upon request.

*Data availability.* Available upon request.

*Author contributions.* Juan-Andrés Pérez-Rúa: Conceptualization, Methodology, Software, Validation, Formal analysis, Data Curation, Writing. Mathias Stolpe: Conceptualization, Methodology, Investigation, Resources, Supervision. Nicolaos A. Cutululis: Validation, Writing, Supervision, Project administration, Funding acquisition.

*Competing interests.* Nicolaos A. Cutululis is a member of the editorial board of Wind Energy Science.

*Acknowledgements.* The research presented in this manuscript has been funded by the Independent Research Fund Denmark (DFF) through the research project *Integrated Design of Offshore Wind Power Plants* with project nr. 1127-00188B

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
