# Peer review of "A Neighborhood Search Integer Programming Approach for Wind Farm Layout Optimization"

_Wind Energy Science, 2022_

## Referee Comment (RC1)

**General Comments**

This work presents a new approach (NSH) to solving the wind farm layout optimization problem using a MILP approach that is made more tractable by a simplified wind farm AEP model. The results of the model and the optimization algorithm are clearly compared to previous works and seem reasonably reproduceable. The work appears to be well-founded from a scientific perspective, is relevant to the subject matter of Wind Energy Science, and provides meaningful contributions to field.

While the work is reasonably well presented, the english grammar and usage in the work present a barrier to understanding. The manuscript should be carefully, preferably professionally, edited to address these concerns so the material will be more accessible, clear, and useful to the community.

**Specific Comments**

**Abstract**

- Line 4-5: "deficit is aimed" I don't know what is meant by this.
- Line 5-6: it is unclear if the heuristic wraps the model (formulations?) or is separate. Consider clarifying.
- Line 8: This sentence was confusing to me, but I think I understand. Consider re-working. I think the intended meaning is that the results of the benchmark problems show that using some substitute objective rather than actual AEP can be a good approach.
- Line 10: "match" is probably a bit strong for the presented results, maybe say the results are competitive or something that does not indicate equality

**1: Introduction**

- Line 17: I don't think I'm convinced about the importance of wind farm layout optimization by this paragraph. You state that wind energy is important politically, is presumably profitable without subsidies, and is a mature industry. The profit and maturity seem to hurt the argument for why this study is important. It sounds like things are just fine without WFLO. I'd suggest re-working this first paragraph. You could consider discussing the tight margins of wind developers and OEMs, especially offshore. You could also mention some hard values for how improved wind farm layouts could reduce the cost of energy even further. Basically, be careful to lay a clear foundation for why this work matters. You don't need to cover a lot of detail or history, but do make a clear case.
- Line 24: I think you are citing Deb (2013) here for an example of a GA, but it reads like you are pointing readers to the GA that Mosetti used, only the dates don't line up (2013 vs 1994). Consider reworking this or putting the expected citation (or no citation, you already cited Mosetti

which presumably has the information on the GA)

- Line 26: Consider removing "and the associated numerical algorithms" because you are stating "main components". Nearly all computational methods will have "associated" algorithms. However, I'd argue that the wake combination model qualifies as a "main" component as well.
- Line 30-34: while the wake model background may not need to be complete, the background given here is not quite correct.
    1. I think Niayifar and Porte-Agel (2015) is mostly focused on the wake combination and turbulence intensity to extend the Bastankhah model to multiple turbines. In this light the citation would be better placed with the wake combination citations. Also note that there is a journal paper by the same authors from 2016 on the topic that may be a better source to cite here.
    2. The Jensen cosine model was actually proposed by Jensen in 1983, so it may be good to cite that paper for the Jensen cosine model as the original source, though the Thomas et al. paper does provide some clarifications.
    3. The list as given seems to show several smooth and differentiable wake models, but the combined citations seem to really only refer to two distinct wake models. I'd suggest making this a little more clear in the discussion.
    4. While the sum of squares or linear combination statement is correct to my knowledge, it may be worth mentioning that the two methods have been used with local and freestream velocity conditions. This makes for four distinct proposed wake combination methods.
        - Linear/freestream: Lissaman 1979
        - Sum of squares/freestream: Katic et al. 1986
        - Linear/local: Niayifar and Porte Agel 2015, 2016
        - Sum of squares/local: Voutsinas 1990 Update: I saw you do discuss this nuance later. It may ok as is, but it did seem incomplete to me at first.
- Line 49: the jump from gradient-based and gradient-free algorithms to discrete algorithms was not clear and needs motivation. Consider stating the connection and purpose of the jump for those unfamiliar with the algorithms ((1) discrete methods are generally a sub-set of gradient-free methods, and (2) why are we talking about them here?)
- Line 59-69: consider also citing https://wes.copernicus.org/articles/7/1137/2022/
- Line 71: how is modeling economic metrics an advantage of the discrete model? This can and has been done in a continuous space for optimization. - see https://onlinelibrary.wiley.com/doi/epdf/10.1002/we.2310
- Line 72: (ii) can be done continuously, but it is more difficult
- Line 72-73: why is (iii) specific to a discrete formulation?
- Line 73-74: what is the distinction between "cost functions" (iv) and "economic metrics" (i)?
- Line 74: fully continuous WFLO has been done with multiple turbine types
    - https://www.wind-energ-sci.net/4/99/2019/wes-4-99-2019.pdf

- Line 76-77: consider elaborating on this idea and why discrete optimization is well suited to overcome the convexity problem

**2:  Physic Modelling**

- Line 97-99: I'm not sure how this statement "No particular . . . " relates to the first sentence in the paragraph. Also, Thomas et al. 2022b specify some restrictions on the mathematical structure for controlling wake diameter and deficit, at least for their purpose. Specifically, the wake deficit and wake diameter must be separately controlled.
- Line 101: from which source? there are two references.
- Line 105: why is Thomas and Ning 2018 cited here and at line 32 for the simplified Gaussian? For the original Bastankhah wake model, I'd suggest citing Bastankhah 2016. For the simplified model, use the citation given in the following sentence (IEA Wind Task 37 2019)
- Line 110: $d_{ij}^{\parallel}$ and $d_{ij}^{\perp}$ are not used in Eq. (1), though the coordinate frame clarifications are helpful, the symbols used seem extraneous at this point in the paper. You could possibly include these symbols as additional equations following the equation explanation of Eqs. (1) and (2) in preparation for use later in the paper.
- Line 115: consider removing one of these duplicate mathematical statements.
- Line 125-130: The references used to arrive at Eqs. (6) and (7) were given in the introduction, but I think it would be helpful to provide them again here.
- Line 133: why is the power curve non-differentiable specifically at rated wind speed? The definition provided in this manuscript is non-differentiable at the rated power, but the continuity of the power curve is just dependent on the power curve definition, so this statement is not correct in general.

**3:  Optimization models**

- Eqs. (12) to (14)
  - The presentation here is difficult to follow. Perhaps consider breaking them up into more equations with more explanation and grouping by interval (1, 2, a+1, m+1, m+2).
  - the statement "for a = 1,. . . ,m" should be applied to each numbered equation it applies to individually.
  - I'm not sure how the delta u is supposed to be applied in Eq. (13). Re-working the presentation of these equations should help.
  - are "a" and "l" being used for the same thing here? If so, correct. If not, please clarify.
  - There may be a better way to present the interval values, the above are just my ideas at the moment.
- Eq. (16a): xi, eta, and u are specified as design variables, but I think eta and u are state variables dependent on xi, so it seems that xi represents

all design variables. I've only seen design variables represented in the sub-scripted variables under the "maximize" in the optimization equation.

- Section 3.2: this approach appears similar to the FLOWERS model found in https://wes.copernicus.org/articles/7/1137/2022/wes-7-1137-2022.pdf. I'd suggest contrasting the method in the submitted manuscript to the FLOWERS model, perhaps in the introduction, but referring back to it again here.
- Line 204: I need a little clarification regarding which "outlook" the IEA 37 studies follow.
- Line 211: please provide justification for dropping the square roots. Why is the model expected to be correct if the square root is simply "dropped"?
- Eq. 20: how did you get to `b_{i,l} + b_{l,i}` and the `l>i`? I don't see offhand how those terms come from combining eqs. 18 and 19 as stated.

**4: Neighborhood search heuristic**

- Alg. 1, Line 13: check spacing
- Line 263-264: what is meant by "stopped until"?
- The NSH algorithm seems similar to the one developed by Paul Malisani and presented in "A Comparison of Eight Optimization Methods Applied to a Wind Farm Layout Optimization Problem" by Thomas et al. (https://wes.copernicus.org/preprints/wes-2022-90/). Consider comparing and contrasting the approaches.

**5: Computational experiments**

- Line 286: why these parameter values?
- Line 299: it would be nice to see all non-default parameters (the introduction "for example" seems to indicate that only some of the non-default parameters are given). Consider putting in a table with the non-default parameter values.
- Line 305-316: was this sampling method compared to any other methods?
- Line 334: how do we know it is "still strong enough"? What was the bar?
- Line 346: which model is "exact"? All the models presented in this paper appear to be approximations.
- Line 349: Perhaps the "deterioration" is partly due to "dropping" the square root?
- Line 349: "this" is unclear, state meaning explicitly
- Table 1, Fig. 4: beautiful use and presentation of correlation. Nice work!
- Line 363: It would be helpful to provide more information about the tuning process.
- Line 363: My understanding of C, T, and V was incomplete and I had to go back and re-read previous sections and this sections to get straitened out. I'd suggest adding more explanation of these inputs when you introduce the algorithm.
- Fig. 5:

- are the times shown clock time or CPU time?
  - while run time is helpful, it can vary drastically depending on implementation, language, system, etc. You may want to consider also including a count of total calls to your objective function.
- Fig. 6, 8:
  - Are your wind turbine markers to scale?
  - This figure is missing axis labels
  - This figure is missing units for the tick labels
- 5.3: the baseline of the percentages given is unclear. Is each percentage given using the last step level as the baseline or the original "incumbant" value?
- Fig. 7: perhaps I missed where this was stated, but are all the AEP values here calculated using the full model for comparison? If not, I think they should be.
- Eq. 23: The equation in your reference is general, but you provide a specific version here. It would be helpful to introduce the general form of your equation from your reference and then fill in the specifics. You may also want to use a more concrete reference here than Investopedia. There are many for this material.
- Line 474: The last sentence here needs more explanation.
- Line 476: I'm not sure what you mean, but if it is the main question then I should. Can you be more specific and/or clarify?
- Fig. 11-13 would probably be more clear if combined and corresponding lines were plotted on the same axes

**Technical Corrections**

**General**

- There are many grammar and usage errors throughout. The manuscript should be carefully edited to address these concerns so the material will be more accessible and useful to the community. I have noted a few of these below.

**Abstract**

**1: Introduction**

- Line 13: "Subsidy-free ..." check grammar
- Line 19: Because you give a list of parts here (rather than just one primary thing), "consists of" may be more appropriate.

**2: Physics modeling**

- Line 132-133: comma after AEP

- Eq. (8): this piece-wise equation contains multiple definitions for some cross-over points. Check the usage of "$<$" vs "$<=$"

**3: Optimization models**

- Line 145: check commas to ensure clarity
- Line 200-203: check grammar and usage to ensure clarity
- Line 205: does "this" refer to the linear superposition model or the power-curve free model? In general, try to avoid "this" where there is any possibility of misinterpretation.

**4: Neighborhood search heuristic**

- Line 243: observation should be singular

**5: Computational experiments**

- Fig. 4: This figure is a little busy, consider giving the figures a little more space by removing all unnecessary elements and adding some buffer space between sub-figures and figure elements. I really like this figure overall though.
- Line 276: radii
- Line 412: I suggest avoiding starting a paragraph with "As shown in Fig. x" because we don't even know what the subject of the paragraph is yet. The "as shown..." should fit well at the end of the sentence.
- Figures in general:
  - The units given sometimes lead to very large numbers that clutter the figure and impede interpretation. I'd suggest using units that reduce the number of digits required in the tick labels (i.e. GWh instead of MWh, and hours or days instead of seconds)

---

## Author Response (AR1)

Wind Energy Science

24th April 2023

Prof. Michael Muskulus
Associate Editor of the Journal,
Editorial Board,
Reviewers

**RESPONSE TO THE EDITOR AND REVIEWERS**

Dear Editor and Reviewers,

We would like to thank you all for your time and insightful comments about our article entitled *"A Neighborhood Search Integer Programming Approach for Wind Farm Layout Optimization"* (submission wes-2022-82), and for considering the topic and proposed method as relevant and promising.

We have made a large effort to improve the quality of the manuscript and to address all the comments and suggestions from both reviewers. The main changes are: (i) restructuring of the introduction, (ii) addition of publications from the literature and several discussions on how they compare/contribute to our work (iii) improvement of most of the figures and tables quality, (iv) reorganization of results section, (v) correction of minor aspects such as typos, writing style and other specific clarifications, and (vi) improvement of the English grammar.

Likewise, we elaborated extensively on concerns from the reviewers about, for example, the contributions from our work regarding the modelling of mixed integer linear programs, and on how the NSH performs an optimization search based on physical information from the wind turbine layouts, and not merely by just following a random search.

Below, each reviewer's comments are approached one by one.

*PLEASE NOTE THAT A REVISED VERSION OF THE MANUSCRIPT WITH CHANGES FROM EACH REVIEWER MARKED WITH RED AND BLUE COLORS IS ATTACHED AT THE END OF THIS LETTER.*

**RESPONSE TO THE REVIEWERS**

**REVIEWER 1**

Find below responses to each of your comments. Modifications are marked in Red:
* * *
*GENERAL COMMENTS*
* * *
*- Comment: This work presents a new approach (NSH) to solving the wind farm layout optimization problem using a MILP approach that is made more tractable by a simplified wind*

Technical University of Denmark
**Department of Wind and Energy Systems**

Frederiksborgvej 399
Building 118
4000 Roskilde
Denmark

Tel  +45 93 51 89 91
Dir. +45 93 51 89 91

juru@dtu.dk
www.vindenergi.dtu.dk

REG-no. DK 30 06 09 46

[Figure]

*farm AEP model. The results of the model and the optimization algorithm are clearly compared to previous works and seem reasonably reproduceable. The work appears to be well-founded from a scientific perspective, is relevant to the subject matter of Wind Energy Science, and provides meaningful contributions to field. While the work is reasonably well presented, the English grammar and usage in the work present a barrier to understanding. The manuscript should be carefully, preferably professionally, edited to address these concerns so the material will be more accessible, clear, and useful to the community.*

- Response:  Thanks for appreciating the contributions presented in the article. Regarding the English grammar and usage, we have conducted a thorough review of it to improve the quality of the manuscript.
* * *
SPECIFIC COMMENTS
* * *
*Abstract*
* * *
- Comment: *Line 4-5: "deficit is aimed" I don't know what is meant by this.*

- Response:  Thanks for pointing out this misleading statement. The word "aimed" has been replaced bv "optimized", so now this sentence should be clearer in transmitting the idea that in the power-curve-free model is optimized a measure closely related to wind speed deficit.
* * *
- Comment: *Line 5-6: it is unclear if the heuristic wraps the model (formulations?) or is separate. Consider clarifying.*

- Response:  Thanks for the suggestion. As an attempt to increase clarity, this is restated as: "A special-purpose neighborhood search heuristic wraps these formulations increasing tractability and effectiveness compared to the full model that is not contained in the heuristic."
* * *
- Comment: *Line 8: This sentence was confusing to me, but I think I understand. Consider reworking. I think the intended meaning is that the results of the benchmark problems show that using some substitute objective rather than actual AEP can be a good approach.*

- Response:  Thanks for the suggestion. As an attempt to increase clarity, this is restated as: "…Numerical results on a set of publicly available benchmark problems indicate that a proxy for total velocity deficit as objective is a functional approach, since high-quality solutions of annual energy production metric are obtained, when using the latter function as substitute objective…"
* * *
- Comment: *Line 10: "match" is probably a bit strong for the presented results, maybe say the results are competitive or something that does not indicate equality.*

- Response:  Thanks for point this out. We agree. This sentence is restated as: "…Furthermore, the proposed heuristic is able to provide good results compared to a large set of distinctive approaches that consider the turbine positions as continuous variables."

[Figure]
* * *
*1: Introduction*
* * *
*- Comment: Line 17: I don't think I'm convinced about the importance of wind farm layout optimization by this paragraph. You state that wind energy is important politically, is presumably profitable without subsidies, and is a mature industry. The profit and maturity seem to hurt the argument for why this study is important. It sounds like things are just fine without WFLO. I'd suggest re-working this first paragraph. You could consider discussing the tight margins of wind developers and OEMs, especially offshore. You could also mention some hard values for how improved wind farm layouts could reduce the cost of energy even further. Basically, be careful to lay a clear foundation for why this work matters. You don't need to cover a lot of detail or history, but do make a clear case.*

*- Response:* Thank you for the remark. Indeed, it may read a bit as a contradiction.

We have re-phrased the paragraph focusing on why lower costs, partially achievable via optimization, are important. This content belongs to the new subsection 1.1 *Motivation and Problem Definition* after recommendation from Reviewer 2.

"…For wind energy to become the cornerstone of a successful green energy transition, further reduction in costs - partly achievable by economically optimized wind farm designs - will play an important role."
* * *
*Comment: Line 24: I think you are citing Deb (2013) here for an example of a GA, but it reads like you are pointing readers to the GA that Mosetti used, only the dates don't line up (2013 vs 1994). Consider reworking this or putting the expected citation (or no citation, you already cited Mosetti which presumably has the information on the GA).*

*- Response:* Thanks for the suggestion. The citation to Deb (2013) has been removed as we agree with this comment.
* * *
*- Comment: Line 26: Consider removing "and the associated numerical algorithms" because you are stating "main components". Nearly all computational methods will have "associated" algorithms. However, I'd argue that the wake combination model qualifies as a "main" component as well.*

*- Response:* Thanks for pointing out this misleading statement. The first sentence of this paragraph is changed to:
"The main components when building an optimization workflow for the WFLO problem are the wake models (deficit and superposition), the program formulation, and the associated numerical algorithms…"
We consider very important to differentiate the three aspects: wake modelling, problem formulation, and numerical algorithms. Essentially, an optimization program is set up by defining each of them according to the needs of the problem and choices of the designer. As

it is well known, there are plethora of wake models that can be used for optimization. Likewise, different problem formulations can be selected, including for example, discrete or continuous modelling, distinctive objective functions and constraints structures, among others. Lastly, it is important to emphasize that for any combination of the previous two components, several solution algorithms can be utilized, for example SLSQP, branch-and-cut, etc. This content belongs to the new subsection 1.2 *Optimization Workflow for WFLO* after recommendation from Reviewer 2.
* * *
*- Comment: Line 30-34: while the wake model background may not need to be complete, the background given here is not quite correct.*

*1. I think Niayifar and Porte-Agel (2015) is mostly focused on the wake combination and turbulence intensity to extend the Bastankhah model to multiple turbines. In this light the citation would be better placed with the wake combination citations. Also note that there is a journal paper by the same authors from 2016 on the topic that may be a better source to cite here.*

*2. The Jensen cosine model was actually proposed by Jensen in 1983, so it may be good to cite that paper for the Jensen cosine model as the original source, though the Thomas et al. paper does provide some clarifications.*

*3. The list as given seems to show several smooth and differentiable wake models, but the combined citations seem to really only refer to two distinct wake models. I'd suggest making this a little more clear in the discussion.*

*4. While the sum of squares or linear combination statement is correct to my knowledge, it may be worth mentioning that the two methods have been used with local and freestream velocity conditions. This makes for four distinct proposed wake combination methods.*
*– Linear/freestream: Lissaman 1979*
*– Sum of squares/freestream: Katic et al. 1986*
*– Linear/local: Niayifar and Porte Agel 2015, 2016*
*– Sum of squares/local: Voutsinas 1990 Update: I saw you do discuss this nuance later. It may ok as is, but it did seem incomplete to me at first.*

*- Response:* Thanks for this really good point. As the reviewer says, the wake model background may not need to be complete, but it should be improved respect to what was presented in the first version. The rest of the paragraph has been edited as

*"…*For formulating tractable frameworks, the designer needs to rely on the so-called engineering wake models. These are essentially mathematical representations which can be expressed in terms of analytical equations after significantly simplifying complex physics modelling, while still capturing to a good extent the underlying nature of the phenomenon under analysis. Scientific articles in this field have proposed and validated engineering wake models with smooth and differentiable velocity deficit shape, two examples are the Bastankhah's Gaussian (Bastankhah and Porté-Agel, 2016) or its simplified version (IEA Wind Task 37, 2019), and the Jensen cosine model

(Jensen, N.O., 1983). Likewise, the aggregation of individual wake velocity deficits can be done through linear superposition (Lissaman, 1979) or root sum squares (Voutsinas et al., 1990), with local or freestream velocity conditions (Porté-Agel et al., 2020)."

We consider that with these modifications the wake modelling state-of-the-art review complies with the points highlighted by the reviewer.
* * *
*- Comment: Line 49: the jump from gradient-based and gradient-free algorithms to discrete algorithms was not clear and needs motivation. Consider stating the connection and purpose of the jump for those unfamiliar with the algorithms ((1) discrete methods are generally a subset of gradient-free methods, and (2) why are we talking about them here?)*

*- Response:* We do not agree with this point, as we believe there are two modelling philosophies with respect to variable types: continuous and discrete optimization. In the subsection 1.3 *Continuous Optimization for WFLO* (please notice that after recommendations from Reviewer 2, the introduction has been split up in subsections), the continuous optimization technique is discussed. For this, both gradient-free and gradient-based algorithms have been utilized in the literature. The paragraph from lines 52 to 57 is new, where the latest work of (LoCascio et al., 2022) is discussed. In subsection 1.4 *Discrete Optimization for WFLO*, the discrete optimization technique is discussed. Within this field, both gradient-free and gradient- based algorithms have also been applied. To clarify this, the following sentence has been added:

*"…Algorithms utilizing explicit gradients are also a valid approach in this field (Pollini, 2022)…"*
This is an article very recently published that adopts a discrete modelling technique as well, using gradient-based algorithms to address the WFLO problem.

The objective of paragraph from lines 69 to 76 is to introduce the motivation of the focus of this manuscript, which is integer programming modelling, presenting the inhering modelling benefits of discrete optimization in this context. Subsection 1.5 *Literature Review for Integer Programming within WFLO* picks up this idea and discusses state-of-the-art in integer programming for the WFLO problem. Lastly, subsection 1.6 *Contributions* discusses the contributions of this manuscript.
* * *
*Comment: Line 59-69: consider also citing https://wes.copernicus.org/articles/7/1137/2022/*

*- Response:* Thanks for giving us notice of this very interesting work. This manuscript is cited in the paragraph from lines 52 to 57. We think that is a great addition to expand the concept of applying simpler objective functions that mitigate the complexity of optimization programs, while still being very competitive finding good solutions compared to more sophisticated models. Thereafter, this article is again cited in lines 96 to 98 to contrast it with the proxy objective function proposed in our manuscript.
* * *
*Comment: Line 71: how is modeling economic metrics an advantage of the discrete model? This can and has been done in a continuous space for optimization. see https://onlinelibrary.wiley.com/doi/epdf/10.1002/we.2310*

*- Response:* In this line it is stated "capacity to include the number of WTs as a variable and to model overall economic metrics as Net Present Value (NPV)". In the mentioned article "Optimization of turbine design in wind farms with multiple hub heights, using exact analytic gradients and structural constraints" the focus is, as the title states, on how to optimize wind farm layouts accounting for WT design. The problem assumes fixed number of WTs. It is clear that the difference lies in the fact that by a discrete modelling technique the number of WTs is considered an optimization variable. With variable number of WTs, the modelling of overall financial metrics as NPV would expose the trade-off between the number of WTs in the farm and the wake losses (AEP) vs investment costs. This is in general not possible in classic continuous optimization frameworks.
* * *
*- Comment*: *Line 72: (ii) can be done continuously, but it is more difficult*

*- Response:* In this line it is stated "…ease of modeling any shape of project area or forbidden zones, convex or non-convex…", meaning exactly what the reviewer says in this comment.
* * *
Comment: *Line 72-73: why is (iii) specific to a discrete formulation?*

*- Response:* In this line it is stated "…capacity to model extensive integrated models to support electrical systems optimization…". The cable layout optimization problem, which designs the electrical network to connect the WTs towards the substations, is a discrete optimization problem. It would be straight-forward to formulate a unified optimization program for the simultaneous wind farm and cable layout optimization problem, if the WFLO is modelled in a discrete way. With both problems being in the same modelling universe, it would be clear which optimization algorithms to explore. If the WFLO is modelled continuously, we cannot envisage a tractable way of tackling the unified problem.
* * *
*- Comment*: *Line 73-74: what is the distinction between "cost functions" (iv) and "economic metrics" (i)?*

*- Response:* An economic metric is defined in this context as the expression used to value a project from the financial perspective. NPV and IRR are examples. Cost functions refer to the mathematical representations to calculate the value of components that are required to fully compute economic metrics.
* * *
*Comment:* *Line 74: fully continuous WFLO has been done with multiple turbine types https://www.wind-energ-sci.net/4/99/2019/wes-4-99-2019.pdf*

*- Response:* We are aware of this. We refer to the *ease* of modelling this aspect. Therefore, the sentence is rewritten as:

"…ease of incorporating multiple WT types, among others…"
* * *
*Comment:* Line 76-77: consider elaborating on this idea and why discrete optimization is well suited to overcome the convexity problem

*- Response:* This is not the meaning of this sentence. The non-convexity nature cannot be overcome. Nevertheless, because of this feature, it is not possible to formally prove optimality. Due to this, usually different solution algorithms will converge into different final solutions. By having a diverse set of available solution algorithms, the likelihood to obtain better solutions for a given problem instance is increased.
* * *
*2: Physics Modelling*
* * *
*Comment:* Line 97-99: I'm not sure how this statement "No particular . . . " relates to the first sentence in the paragraph. Also, Thomas et al. 2022b specify some restrictions on the mathematical structure for controlling wake diameter and deficit, at least for their purpose. Specifically, the wake deficit and wake diameter must be separately controlled

*- Response:* Thanks for the feedback. To improve readability and connection between sentences, the statements have been rewritten like this

"The proposed MILP models and general optimization framework in this article can be easily applied to many wake deficit models. No particular properties on smoothness or differentiability are required from these models for optimization purposes. Additionally, no specific demands on mathematical structure in connection with controlling wake diameter and deficit (Thomas et al., 2022b) are stemming from the optimization programs proposed in this article…"
* * *
*Comment:* Line 101: from which source? there are two references

*- Response:* Thanks for pointing out this redundancy on the references. Reference (Dykes et al., 2015) has been deleted, because (Baker et al., 2019) is an indexed paper containing the information about wake model and benchmark results.
* * *
*- Comment:* Line 105: why is Thomas and Ning 2018 cited here and at line 32 for the simplified Gaussian? For the original Bastankhah wake model, I'd suggest citing Bastankhah 2016. For the simplified model, use the citation given in the following sentence (IEA Wind Task 37 2019)

*- Response:* Thanks for the suggestion. For the original wake model, it is cited (Bastankhah and Porté-Agel, 2016) and for the simplified model (IEA Wind Task 37, 2019), as suggested by the reviewer.
* * *
*- Comment:* Line 110: $d_{ij}^{\parallel}$ and $d_{ij}^{\perp}$ are not used in Eq. (1), though the coordinate frame clarifications are helpful, the symbols used seem extraneous at this point in the paper. You could possibly include these symbols as additional equations following the equation explanation of Eqs. (1) and (2) in preparation for use later in the paper.

- *Response:* Thanks for the suggestion. We consider that defining $d_{ij}^{\parallel}$ and $d_{ij}^{\perp}$ is useful at this point of the paper because variables $\bar{x}_{\ell}$, $\bar{x}_i$, $\bar{y}_{\ell}$ and $\bar{y}_i$ are introduced here. Trying to improve readability, this paragraph is restated as
"where $u_{\infty}$ is the inflow wind speed, $C_{\text{T}}$ is the thrust coefficient, $\bar{x}_i - \bar{x}_{\ell}$ is the streamwise distance from the hub generating wake ($\bar{x}_{\ell}$) to hub of interest ($\bar{x}_i$) along freestream (let this difference be $d_{i\ell}^{\parallel}$), $\bar{y}_i - \bar{y}_{\ell}$ is the span-wise distance from the hub generating wake to hub of interest perpendicular to freestream (let this difference be $d_{i\ell}^{\perp}$), $\sigma_{\text{y}}$ is the standard deviation of the wake deficit, $k_{\text{y}}$ is a variable based on a turbulence intensity, and $D$ is the WT diameter."
* * *
- *Comment*: Line 115: consider removing one of these duplicate mathematical statements.

- *Response:* Thanks for noticing this typo. It has been fixed.
* * *
- *Comment*: Line 125-130: The references used to arrive at Eqs. (6) and (7) were given in the introduction, but I think it would be helpful to provide them again here.

- *Response:* Thanks again for this recommendation. The references have been added above Eq. (3), Eq. (4), Eq. (6), and Eq. (7).
* * *
*Comment*: Line 133: why is the power curve non-differentiable specifically at rated wind speed? The definition provided in this manuscript is non-differentiable at the rated power, but the continuity of the power curve is just dependent on the power curve definition, so this statement is not correct in general.

- *Response:* We have modified this subsection as follows to improve the technical rigor
"Suitable power curves are required for computing AEP. Often, power curves are not perfectly suitable for optimization, due to the usual non-differentiability in several points throughout the function. Generally, a power curve is zero below cut-in wind speed, zero above the cut-out wind speed, and constant between the rated wind speed and the cut-out wind speed. In this particular study, between the cut-in and rated wind speeds the curve is assumed to be smooth, convex and monotonically increasing. The simplified power curve for a generic turbine as a function of wind speed u is modelled through…"

"…where $p^{rated}$ is the nominal power at (and above) rated wind speed $u^{rated}$. The other turbine characteristics are the cut-in wind speed $u^{cut-in}$, and the cut-out wind speed $u^{cut-out}$. In this definition, the WT power curve is not differentiable at $u^{cut-in}$, $u^{rated}$, $u^{cut-out}$, since in these points the left and right hand side derivatives are different. Be aware that the optimization programs proposed in this manuscript are not dependent on WT power curve differentiability." The non-differentiability discussed here is naturally dependent on the power curve definition. However, the one presented aligns with the usual function recurrently implemented in the literature.
* * *
*3: Optimization Models*
* * *
*- Comment: Eqs. (12) to (14) The presentation here is difficult to follow. Perhaps consider breaking them up into more equations with more explanation and grouping by interval (1, 2, a+1, m+1, m+2).*

*– the statement "for a = 1,. . . ,m" should be applied to each numbered equation it applies to individually.*

*– I'm not sure how the delta u is supposed to be applied in Eq. (13). Re-working the presentation of these equations should help.*

*– are "a" and "l" being used for the same thing here? If so, correct. If not, please clarify.*

*– There may be a better way to present the interval values, the above are just my ideas at the moment.*

*- Response*: Thanks for this observation. This part has been reworked as presented in the next page.

We hope that this improves the readability of this part. The statement "for a = 1,. . . ,m" has been applied individually in Eq. (14) and in Eq. (15), meaning each of the subintervals sampled within the cubic subdomain of the whole WT power curve domain. The value of $\Delta u$ has been explicitly declared in the paragraph preceding the equations. Lastly, $a$ and $l$ represents different things. The variable $l$ is for any interval in the whole domain of the curve, while $a$ is for an interval located in the cubic domain.

"

cut-out speed. Each isometric interval within the cubic domain of length $\Delta u = \left(u^{\text{rated}} - u^{\text{cut-in}}\right)/m$, is approximated with a constant power value, see Fig. 1.

An interval $l$ of the whole domain is characterized by three parameters $u_{\text{s}}^l$, $u_{\text{m}}^l$, and $u_{\text{h}}^l$ with the next properties

$$u_{\text{s}}^1 = -u^{\text{ini}}, u_{\text{h}}^1 = u^{\text{cut-in}}, u_{\text{s}}^{m+2} = u^{\text{rated}}, u_{\text{h}}^{m+2} = u^{\text{cut-out}} \tag{12}$$

$$u_{\text{s}}^2 = u^{\text{cut-in}}, u_{\text{h}}^{m+1} = u^{\text{rated}} \tag{13}$$

$$u_{\text{s}}^{a+1} = u^{\text{cut-in}} + (a-1)\Delta u \text{ for } a = 1,\ldots,m \tag{14}$$

$$u_{\text{h}}^{a+1} = u^{\text{cut-in}} + a\Delta u \text{ for } a = 1,\ldots,m \tag{15}$$

$$u_{\text{m}}^l = 0.5(u_{\text{s}}^l + u_{\text{h}}^l) \tag{16}$$

"

Equation (12) defines the lower and upper limits for the extreme intervals $l = 1$ and $l = m + 2$, Eq. (13) formalizes the lower and upper limits for the first interval in the cubic part, $a = 1$, and the last one $a = m$, respectively. Equation (14) expresses the lower limits for intervals in

the cubic part ($a = 1, \ldots, m$), while Eq. (15) does it for the upper limits. Equation (16) presents how to determine the extracted wind speed associated to the interval $l$ of the whole domain, which is the average value of $u_s^l$ and $u_h^l$."
* * *
- *Comment: Eq. (16a): xi, eta, and u are specified as design variables, but I think eta and u are state variables dependent on xi, so it seems that xi represents all design variables. I've only seen design variables represented in the sub-scripted variables under the "maximize" in the optimization equation.*

- *Response*: The reviewer is right about the fact that eta and u are state variables fully dependent on xi. However, we do not agree with the observation that the subscripted variables under "maximize" should only present fully independent variables. For completeness, we chose to present all variables required in an optimization program, regardless of relation of dependence between them. On the other hand, the article explains clearly the difference between binary variables \xi and the state ones eta, u, and tau.
* * *
- *Comment: Section 3.2: this approach appears similar to the FLOWERS model found in https://wes.copernicus.org/articles/7/1137/2022/wes-7-1137-2022.pdf. I'd suggest contrasting the method in the submitted manuscript to the FLOWERS model, perhaps in the introduction, but referring back to it again here.*

- *Response*: This manuscript is cited in the paragraph from lines 52 to 57. Thereafter, this article is again cited in lines 96 to 98 to contrast it with the proxy objective function proposed in our manuscript. Finally, this article is referred back in this section in lines 269 to 271 as "This proxy objective function is very useful for formulating the program in the MILP category. While the work in (LoCascio et al., 2022) focuses on a different formulation (likely more accurate analytically than the one presented here) that is non-linear but gradient friendly, hence useful for continuous gradient-based optimization."
* * *
- *Comment: Line 204: I need a little clarification regarding which "outlook" the IEA 37 studies follow.*

- *Response*: We have rewritten this sentence to improve readability as follows
"Albeit the formulation of Sect. 3.1 represents to a very large extent the physics ruling the problem, it has a considerable number of variables and constraints that may hinder the capacity to tackle larger problems. The model presented in this section neglects power curve and AEP calculation and aims at simplifying the power-curve-based version.

The model deploys a strategy to account for the combination of Eq. (3) and Eq. (7) to calculate velocities, since the case studies from the IEA Wind Task 37 follow this methodology for AEP computation. It would be possible though to consider the linear superposition model if necessary. However, the power-curve-free model does not support the application of Eq. (4)."

We mean that that approach is considered in this section for AEP computation, following the methodology implemented in the IEA37 Wind Task.
* * *
- *Comment: Line 211: please provide justification for dropping the square roots. Why is the model expected to be correct if the square root is simply "dropped"?*

*- Response*: We acknowledge that by simply dropping the square roots the model is not "correct", but expect that the resultant expression, incorporated in the MILP model, is "*good enough*". Line 244 in the new version of the manuscript is added:

"…the arguments of the square roots in Eq. (19) define a function closely related to the full root-squared expression…"

To have a better idea about this premise, please see the below plot

[Figure]

The plane plot is of the function $z = x + y$ and the other one is for $z = \sqrt{x} + \sqrt{y}$. Note how these two functions follow relatively close to each other for non-negative values of $x$ and $y$. A similar behavior is expected between the original root-squared expression and the other one that ignores them.

Practical evidence of the accuracy of this simplification is presented in Table 1.
* * *
*- Comment: Eq. 20: how did you get to b_{i,l} + b_{l,i} and the l>i? I don't see offhand how those terms come from combining eqs. 18 and 19 as stated.*

*- Response*: As variable $z_{i\ell}$ represents that both WTs in $i$ and $j$ are selected, then when it is zero, the mutual influence given by summing up both $b_{i\ell}$ and $b_{\ell i}$ must also be zero. By defining the second sum with $\ell > i$, the number of variables is halved after this symmetric property.
* * *
*4: Neighborhood Search Heuristic*
* * *
*- Comment: Alg. 1, Line 13: check spacing*

*- Response*: Thanks for noticing this typo. It has been fixed.
* * *
*- Comment: Line 263-264: what is meant by "stopped until"?*

*- Response*: Thanks for noticing this misleading statement. It has been restated as "…The complete model is sent to the MILP solver with ξ as warm-starter, stopped when reaches either optimality or the assigned maximum computing time $T$…"
* * *
*- Comment: The NSH algorithm seems similar to the one developed by Paul Malisani and presented in "A Comparison of Eight Optimization Methods Applied to a Wind Farm La out*

*Optimization Problem" by Thomas et al. (https://wes.copernicus.org/preprints/wes-2022-90/). Consider comparing and contrasting the approaches.*

- *Response*: Thanks for giving us notice of this very interesting work. We have added this work in the introduction (paragraph from lines 99 to 109) as

"The second main contribution is the proposition of a new special purpose neighborhood search heuristics in order to speed up the generation of high-quality solutions. This heuristic, wrapping both formulations, has a twofold functionality; first to increase tractability, and second to redirect the optimization search in terms of a specified objective function with higher fidelity. Similar neighborhood search methods have been proposed in the literature, as the Discrete exploration-based optimization (DEBO) (Thomas et al., 2022c), which is a two-steps process composed by a greedy initialization and a local search block. While the method proposed in this manuscript shares most of the advantages of the mentioned approach (no gradients required, can handle unconnected and non-convex boundary constraints, and so on), it actually goes beyond the DEBO algorithm as among others, i) significantly less AEP function evaluations are required, and ii) it is based on well-establish integer programming theory, relying in efficient implementations of the branch-and-cut algorithm. The main numerical results indicate good computational performances for a set of publicly available  benchmark case studies compared to state-of-the-art gradient-free and gradient-based approaches (Baker et al., 2019)."

This article is referred back in this section in paragraph from lines 305 to 307 as

"… One of the advantages of the NSH compared to the DEBO algorithm (Thomas et al., 2022c) is the reduced number of AEP evaluations. In an iteration κ, only |S| evaluations are required. Likewise, many of the other expensive calculations are done in a preprocessing stage…"

"…Another difference between the NSH and the DEBO is that the latter only changes the position of a single WT in a given iteration, while the former considers simultaneous modifications of several WT positions." (lines 312 to 314).
* * *
*5: Computational Experiments*
* * *
- *Comment:  Line 286: why these parameter values?*

- *Response*: In the line 320 has been added the reference for benchmarking
"The main parameters of the wake model in Sect. 2.1 are fixed to CT = 8/9 and ky = 0.0324555, according to (Baker et al., 2019)"
* * *
- *Comment:  Line 299: it would be nice to see all non-default parameters (the introduction "for example" seems to indicate that only some of the non-default parameters are given). Consider putting in a table with the non-default parameter values.*

- Response: This paragraph has been modified to
"…The selected MILP solver is the commercial branch-and-cut algorithm implemented in IBM ILOG CPLEX Optimization Studio V20.1 (IBM, 2022). Apart from the number of threads and time limit settings, a few other parameters are also set to different values compared to the

default choices as well. One is the parameter returning high-quality feasible solutions early in the process, for which, the (CPX_MIPEMPHASIS_HEURISTIC) is activated. The intention is to generate more feasible layouts which is important for the neighborhood search algorithm. Additionally, strong branching is used for variable selection given the large size of the models (CPX_VARSEL_STRONG is selected). The intention is to reduce the size of the search tree and thus the memory requirements compared to default settings."

Since these are the only settings that have been changed from default values, we choose not to add a table for this purpose to avoid enlarging the paper's length.
* * *
*- Comment:  Line 305-316: was this sampling method compared to any other methods?*

- Response: Not being the objective of the manuscript to evaluate different sampling methods, this has not been exhaustively investigate. One of the experiments not included in the article was to use a Delaunay-triangulation-based sampling of the 1300 m radius circumference. Using the same algorithm parameters, the presented method in the manuscript consistently improved the Delaunay one. Because more experiments should be done to elaborate a comprehensive comparison, no discussion is presented in this matter. This could be an interesting support work to perform in the short-term future.
* * *
*- Comment: Line 334: how do we know it is "still strong enough"? What was the bar?*

- Response: Thanks for the feedback. We agree with the fact that this expression may sound as comparative to a well-defined standard. Instead, this sentence has been modified to
"..In spite of this deterioration, the linear correlation is still considered quite strong.."
Although the range of correlation coefficient values and the corresponding levels of correlation vary depending on the application context, a correlation in the interval [-1 to -0.80] is usually deemed as 'Very Strong Negative', and between [-0.79 to -0.60] as 'Strong Negative'. See for example reference https://www.ccsenet.org/journal/index.php/cis/article/view/59661.
Since we do not aim to provide a formal definition of this aspect, adding the word "considered" should highlight the subjective meaning intended.
* * *
*- Comment: Line 346: which model is "exact"? All the models presented in this paper appear to be approximations.*

- Response: This comment refers to the general finding of the article that focusing on total wind speed minimization (or its use to calculate an approximated AEP function) is a promising research line for the WFLO problem.
* * *
*- Comment: Line 349: Perhaps the "deterioration" is partly due to "dropping" the square root?*

- Response: This is true and it is actually discussed in the lines 383 to 390.
* * *
*- Comment: Line 349: "this" is unclear, state meaning explicitly*

- Response: It has been replaced "this" by "…Case III…".
* * *
*- Comment: Table 1, Fig. 4: beautiful use and presentation of correlation. Nice work!*

- Response: Thanks!
* * *
- *Comment: Line 363: It would be helpful to provide more information about the tuning process.*

- Response: Since we consider that there is not an optimal way of tuning these parameters, no extensive discussions are deployed. These settings were obtained after few trial and error. Some general annotations are given in the following lines about the reasoning behind the presented values of $C$, $T$, and $V$.
* * *
- *Comment: Line 363: My understanding of C, T, and V was incomplete and I had to go back and re-read previous sections and this sections to get straitened out. I'd suggest adding more explanation of these inputs when you introduce the algorithm.*

- Response: Done. This is carried out by
"..The main inputs are C = {467, 590, 1014} (set of candidate locations), T = {1, 1.5, 2} h (set of max computing times for each candidate location), V = {2, 4, 6, 16} (set of neighborhood search sizes). See Sect. 4..."
* * *
- *Comment: Fig. 5: – are the times shown clock time or CPU time? – while run time is helpful, it can vary drastically depending on implementation, language, system, etc. You may want to consider also including a count of total calls to your objective function.*

- Response: It is indeed clock time and it has been clarified in Line 392.

To our knowledge, function evaluation metric is usually used to assess metaheuristic algorithm's performance as they depend upon the number of generations and the size of the population, so it is an indication of the efficiency of the algorithm, considering a given computing time to assess the fitness function once. Some gradient-based solver also provide this metric. However, the proposed method uses an exact formulation and calls an external state-of-the-art solver using branch-and-cut method to get high-quality solutions. We see that the vast majority of works in the operations research field using solvers as CPLEX report clock time as normal practice. See for example https://link.springer.com/article/10.1007/s10732-015-9295-0#:~:text=Relax%2Dand%2Dfix%20(RF,in%20their%20sophisticated%20lot%2Dsizing, or https://www.sciencedirect.com/science/article/pii/S2211692317300188.
This is usually the case because branch-and-cut black-box solvers do not easily provide this information.
* * *
- *Comment: Fig. 6, 8: – Are your wind turbine markers to scale? – This figure is missing axis labels – This figure is missing units for the tick labels*

- Response: Markers are not to scale. Figures 3, 6, and 8 have been edited so axis labels and units for ticks are added.
* * *
- *Comment: 5.3: the baseline of the percentages given is unclear. Is each percentage given using the last step level as the baseline or the original "incumbant" value?*

[Figure]

- Response: The baseline is the last step commented. In line 449-450 has been added the sentence

"…As for Case I, improvement percentages are calculated using the last commented step as the baseline…"
* * *
- *Comment*: *Fig. 7: perhaps I missed where this was stated, but are all the AEP values here calculated using the full model for comparison? If not, I think they should be.*

- Response: Correct. This is stated in line 320-326.
* * *
- *Comment*: *Eq. 23: The equation in your reference is general, but you provide a specific version here. It would be helpful to introduce the general form of your equation from your reference and then fill in the specifics. You may also want to use a more concrete reference here than Investopedia. There are many for this material.*

- Response: From lines 502 to 507, the following description has been added

"…. The general form of the NPV equation (Cogency, 2014) is defined by the sum of the present value of cash flows (Discounted Cash Flow, DCF) of a project under analysis. In Eq. (25), the first sum is a negative cash flow representing purchase of the WTs at the construction stage of the project, while the next term represents positive cash flows coming from trading the electricity in the market. Because of the additive nature of the NPV metric and since the focus is on evaluating investment vs revenues, by maximizing Eq. (25), a fully comprehensive NPV metric is equivalently improved."
* * *
- *Comment*: *Line 474: The last sentence here needs more explanation.*

- Response: By expanding the previous paragraph and with the following sentence, we consider that the explanation has been improved

"When the number of turbines is fixed to 10, the NPV evolution (green line in Fig. 11b) is driven by the AEP (green line in Fig. 11a). Both curves are monotonically increasing, reaching a final value of NPV of = 456.40 mEUR. The same behaviour is visible for $n_T$ = 50, although the final NPV is greater (683.53 mEUR), see blue line Figure 11b. In the second study, the positive difference in DCF from the revenues surpasses the associated extra investment costs from the additional 40 wind turbines considered. The significant increase in the number of WTs doubles the computing time, due to the large increase in the number of variables, selecting 50 WTs entails significantly more possible combinations of valid solutions."
* * *
- *Comment*: *Line 476: I'm not sure what you mean, but if it is the main question then I should. Can you be more specific and/or clarify?*

- Response: The sentence is restated as

"…An interesting question is whether there is a larger NPV in between the bounds of WT number…."
* * *
- *Comment*: *Fig. 11-13 would probably be more clear if combined and corresponding lines were plotted on the same axes*

- Response: Thanks for the advice. The recommendation has been followed and the descriptive texts have correspondingly been adapted.
* * *
*TECHNICAL CORRECTIONS*
* * *
*General*
* * *
- *Comment*: *There are many grammar and usage errors throughout. The manuscript should be carefully edited to address these concerns so the material will be more accessible and useful to the community. I have noted a few of these below.*

- Response: We have conducted a thorough review of it to improve the quality of the manuscript.
* * *
*1: Introduction*
* * *
- *Comment*: *Line 13: "Subsidy-free . . . " check grammar*

- Response: Done.
* * *
- *Comment*: *Line 19: Because you give a list of parts here (rather than just one primary thing), "consists of" may be more appropriate.*

- Response: Done. Thanks for the advice.
* * *
*2: Physics Modelling*
* * *
- *Comment*: *Line 132-133: comma after AEP*

- Response: Done. Thanks for the advice.
* * *
- *Comment*: *Eq. (8): this piece-wise equation contains multiple definitions for some cross-over points. Check the usage of "<" vs "<="*

- Response: Thanks for noticing this typo. We had seen it in advance, and it has been corrected.
* * *
*3: Optimization Models*
* * *
- *Comment: Line 145: check commas to ensure clarity*

- Response: Done. Thanks for the advice.
* * *
- *Comment: Line 200-203: check grammar and usage to ensure clarity*

- Response: Done. See full paragraph.

…Albeit the formulation of Sect. 3.1 represents to a very large extent the physics ruling the problem, it has a considerable number of variables and constraints that may hinder the capacity to tackle larger problems. The model presented in this section neglects power curve and AEP calculation and aims at simplifying the power-curve-based version."
* * *
- *Comment: Line 205: does "this" refer to the linear superposition model or the powercurve free model? In general, try to avoid "this" where there is any possibility of misinterpretation.*

- *Response:* Thanks for the advice. Corrected as

"…However, the power-curve-free model does not support the application of Eq. (4)…"
* * *
*4: Neighborhood Search Heuristic*
* * *
- *Comment: Line 243: observation should be singular*

- Response: Thanks for noticing this typo. It has been corrected.
* * *
*5: Computational Experiments*
* * *
- *Comment: Fig. 4: This figure is a little busy, consider giving the figures a little more space by removing all unnecessary elements and adding some buffer space between sub-figures and figure elements. I really like this figure overall though.*

- *Response:* Thanks for the advice. The AEP units in this figure has been changed to GWh and a buffer space between the top sub-figures has been added as well.
* * *
- *Comment: Line 276: radii*

- Response: Thanks for noticing this typo. It has been corrected.
* * *
- *Comment: Line 412: I suggest avoiding starting a paragraph with "As shown in Fig. x" because we don't even know what the subject of the paragraph is yet. The "as shown. . . " should fit well at the end of the sentence.*

- Response: Thanks for the advice. It has been corrected and checked throughout the manuscript.
* * *
- *Comment: Figures in general: – The units given sometimes lead to very large numbers that clutter the figure and impede interpretation. I'd suggest using units that reduce the number of digits required in the tick labels (i.e. GWh instead of MWh, and hours or days instead of seconds)*

- Response: For Figure 4 and Figure 11 this comment is particularly useful and it has been applied. For Figures 5, 7, and 9 the ordinate units has been changed to GWh. The abscissa units (s) has been kept according to the needs of the descriptive text.

[Figure]

**REVIEWER 2**

Find below responses to each of your comments. Modifications are marked in Blue:
* * *
**REVIEWER SUMMARY**
* * *
*- Comment: The authors consider the wind farm layout optimization problem. Their contribution are a pair of discrete optimization algorithms. These consist of an inner and outer optimizer. Starting from an initial proposal solution (feasible layout) The inner optimizer generates a set of candidate solutions (feasible layouts; 'pool') using a MILP solver applied to a linearization—thus approximation—of the classical AEP objective. (The authors propose two such linearizations, hence the pair of algorithms.) The outer 'NSH' optimizer calculates the exact, non-approximated AEP values for the candidate solutions and selects the best one as the proposal for the next MILP run (or the final solution, upon algorithm termination) and determines the parameters for the next MILP run.*

*The authors motivate their work by pointing out the advantages of discrete optimization algorithms when moving beyond classical AEP calculations to objectives such as NPV which also take into account costs, such as those of turbines, their installation, and the cabling. In support of the linearization of the objective—which is needed to use well-developed, capable MILP solvers—they present an analysis of the correlation between the AEP and the simplest linearized objective based on their application to a set of random layouts. To demonstrate their algorithms, they apply them to the IEA Wind Task 37 Case Study 1, which consists of three layout optimization problems with a disc-shaped site with different sizes and (fixed) turbine numbers. Furthermore, they also modify the Case Study problem, to show that one of the two algorithms can optimize NPV by varying both turbine locations and counts.*
* * *
**GENERAL EVALUATION AND KEY CRITICAL FEEDBACK POINTS**
* * *
*- Comment: I was able to follow the exposition without much rereading, so generally I consider the paper to be written at good levels of abstraction and detail and well-structured. The quality of the writing is decent: while the meaning is generally clear, the reader is distracted by some strange formulations, likely due to the authors not being native speakers of English. This can be fixed by having a (near) native speaker going over the paper just focusing and providing feedback on English usage (some pointers will be given in the detailed feedback). The paper's visuals and tables are generally OK, but can be improved in some detailed aspects (pointers will be provided in the detailed feedback). The mathematics as well is generally fine, but the notation can be introduced with a bit more care to support understanding and avoid confusion (pointers will be provided in the detailed feedback).*

*I agree with the paper's motivation for developing a discrete wind farm layout optimization algorithm, but did not find the argumentation sufficiently concrete or precise. Make it more explicit what and how is difficult to do with continuous optimization algorithms and easy with discrete ones. A weakness of the paper is that it only demonstrates the advantages of discrete optimization by looking at one optimization problem that goes beyond the basic AEP optimization over a convex site only by making the turbine count a design variable. Furthermore, the choice of benchmark used, IEA Wind Task 37 Case Study 1 is not anymore*

state-of-the art; something like Case Study 4, with a more realistic site complexity and more realistic wind resource modeling would be a more appropriate comparison point (cf. the paper on this Case Study currently also under review: https://wes.copernicus.org/preprints/wes-2022-90/ [disclosure: I contributed to this work]). Ideally, such a more realistic benchmark would be included, but this may be too much to ask, so I would leave such decision to the editor.

The correlation analysis to support the use of one of the proposed linearized objectives is useful to show that it makes sense to try out its use, as indeed there is sufficiently strong correlation within a random set of layouts. However, I find correlation for a random set of layouts insufficient justification for concluding that its use is warranted. Namely, during a MILP run, the layouts will very much not be random and will be very similar to each other, so it would need to be shown that for such sets of layouts there also is correlation. This can be done by comparing the rankings produced by the linearized objective and the AEP for all of the solutions in a pool, for each of the pools. It is my opinion that such an analysis should be added, because based on the current information, it could still be the case that the inner MILP optimization's added value is limited, as there is mostly a random search happening. (It the plots 5, 7 and 9, the flat parts suggest such behavior and the sloped ones not.)

Mathematical programming techniques (linear, quadratic, with and without integer variables) have been used in the context of wind farm layout optimization by many researchers. Also, linearization of objectives to be able to use MILP solvers is a common technique. The paper gives a decent selection of references about this. It can make the connection to the existing literature more explicit, however. Namely, what is it precisely that is new in this work, which was not done yet (in this context)? Specifically, I feel that the work of Turner et al. (2014) should get more credit and mention in your discussion, as they present a linearization that is very close to your power curve-free approach (compare your Eqs. 17-19 with their Eqs. 3, 7, 9 and your discussion on page 9 line 219-220 with their Eq. 10). As far as I can see, your contribution here is applying a big-M trick, which reduces the problem's complexity.

Similarly, a bit more can be said about search heuristics both in wind farm layout optimization and other application domains to contextualize your NSH algorithm. (Such search algorithms are quite common, also in nested optimization algorithms.) Here, I am not aware of as close a counterpart as with Turner et al. (2014) for the MILP part, but some example references are http://dx.doi.org/10.1016/j.renene.2015.01.005 (the most important one, according to me), http://dx.doi.org/10.1016/j.renene.2012.07.021, and methods in https://wes.copernicus.org/preprints/wes-2022-90/ such as ADREMOG [disclosure: I was involved in the creation of this algorithm].

The discussion of the application of the presented algorithms to IEA Wind Task 37 Case Study 1 presents information about their practical computational efficiency. The computational time needed shows the approach to be very computationally demanding (order of 10 h, 20 h, 40 h, respectively for the sites with 16, 36, and 64 turbines), which impedes its use and therefore decreases its practical relevance. Certainly the variant using a stepwise-constant approximation to the power curve is problematic in this regard, as on very performant hardware it takes 36 hours on the smallest case to perform noticably worse than the power curve-free variant (using about 15 h on quite standard workstation laptop hardware). In the IEA Wind Task Case Studies, there was a wide range of computational efficiencies in the algorithms presented, but there were multiple competitive ones that were substantially more efficient than

*the approaches presented here. As an example, the approach that I contributed to the Case Studies is the one based on pseudo gradients (cf. https://wes.copernicus.org/articles/6/815/2021/). The authors mention the results of this approach with their own by stating that they can achieve similar results in 2 and 3 hours (16 and 36 turbine cases) and that this is 'way faster' than for 'these kind of algorithms'. I re-ran the case studies with the (current, more flexible, but more involved) version of the pseudo-gradient algorithm and obtained the following results (on a laptop that is roughly 2-3 times as fast as the one used by the authors), to be compared to the values in Table 1:*

- *16-turbine case: 4 seconds to obtain 403 GW layout; 1 minute to obtain 409 GW layout*

- *36-turbine case: 7 seconds to obtain 833 GW layout; 2 minutes to obtain 844 GW layout*

- *64-turbine case: 45 seconds to obtain 1466 GW layout; 2 minutes to obtain 1480 GW layout*

*While the pseudo-gradient algorithm was created for supporting a more interactive form of wind farm design and therefore is not able to obtain the highest AEP values seen, these computation times should help recalibrate the author's view of what is typical in state-of-the-art wind farm layout optimization. Understandably, I feel it is required that the authors update their mention of typical times of approaches they compare to.*

*- Response:* Thanks for your feedback. We respond to each of the comments contained in this section as follows:

- English usage: We have thoroughly checked the English grammar throughout the manuscript to improve its quality.
- Paper's motivation for developing a discrete wind farm layout optimization algorithm: We have modified the whole introduction section. This point should be clearer now. With discreet optimization, when the problem consists of complexly shaped wind farm areas, as a set defined by disconnected non-convex polygons, simply generating a set of candidate locations within these areas would suffice to satisfy the domain constraint. While we have no direct experience in applying continuous gradient-based algorithms for this case, the available literature indicates increased complexity, since functions based on minimum distance to the set of vertices of a polygon, and customized functions to evaluate if a point is inside or not of that shape, must be incorporated. The full effect of using these functions in the context of gradient-based optimization does not seem to be known. However, one expects that due to the non-differentiability of the functions throughout the whole domain, the performance of numerical algorithms will be impacted.

  Other advantages of discrete optimization approach are the ability to easily consider a variable number of installed wind turbines, or more interestingly for us, capacity for unified optimization with electrical systems (cable layout). To the best of our knowledge, no implementations using continuous optimization are available for the unified problem.

  The Case IV in paper *https://wes.copernicus.org/preprints/wes-2022-90/* was not available to us at the development stage of the methods and during the preparation of the first version of the manuscript. As the reviewer argues, for the presented cases (Cases I, II, and III from the IEA first benchmark), some of the benchmark algorithms (gradient-based or pseudo-gradients) are computationally faster than our proposed

methods. Nevertheless, the main aim of the manuscript is to provide a proof of concept of the methods. While evidently slower, the comparison is aimed at showing that the results obtained by the proposed methods are in the same quality range as the benchmark.

In the paper *https://wes.copernicus.org/preprints/wes-2022-90/* the best results are obtained after applying the DEBO algorithm, which follows a discretized modelling technique as well. This should be understood as an advantage of our proposed methods, showing the promising potential of the method for instantiations of the WFLO problem beyond classic definition.

- Correlation proxy total velocity deficit vs AEP: Based on the reviewer's suggestion, we have extended the plot from Fig. 4c by including all solutions from each solution pool when executing the NSH using model (23) (total wind speed deficit proxy) for Cases I, II, and III. The results for Case I clearly show that the layouts are not random and are similar to each other in a most of the iterations within the NSH. See Fig. A. While it is true that in a solution pool in a given iteration the correlation can be worse (for example, in the box of [0.30 m/s-0.39 m/s,407 GWh-414 GWh], see Fig. B), this negative effect is corrected by the NSH (line 13 in Algorithm 1 in the manuscript), by choosing as new incumbent for next iteration the best solution in terms of AEP. See Fig. C. We see that the frequency of occurrence of low correlation within the solution pool in a specific NSH iteration is limited.

[Figure]

**Figure A. Correlation plot AEP vs Linear objective for all solutions in all pools for Case I (total number of feasible points of 229)**

[Figure]

**Figure B. Correlation plot AEP vs Linear objective for solutions in a single pool for Case I**

[Figure]

**Figure C. Correlation plot AEP vs Linear objective after NSH correction for Case I**

As presented in Table I, we see that the correlations deteriorate for the larger case studies. This is also the situation in the solution pool analysis, as illustrated in Fig. D to Fig. G for the 36 and 64 WTs cases. The solution pools with a low correlation are marked with red circles in Fig. D and Fig. F. Correspondingly, Fig. E and Fig. G show how the NSH helps in improving the correlation metrics from the point of view of incumbent generation, by recentering the optimization search around a new point in an iteration $\kappa + 1$, only if it improves the best-known solution at $\kappa$. The total number of feasible points in each case after summing up in all pools is relatively low: Case I - 229 valid WT layouts generated throughout the entire optimization search (and 412 and 714, respectively, for the next two cases). With a random search, the number of valid

[Figure]

WT layouts would, intuitively, be significantly larger.

[Figure]

**Figure D. Correlation plot AEP vs Linear objective for all solutions in all pools for Case II (total number of feasible points of 412)**

[Figure]

**Figure E. Correlation plot AEP vs Linear objective after NSH correction for Case II**

[Figure]

**Figure F. Correlation plot AEP vs Linear objective for all solutions in all pools for Case III (total number of feasible points of 714)**

[Figure]

**Figure G. Correlation plot AEP vs Linear objective after NSH correction for Case III**

The flat parts of Fig. 5, Fig. 7, and Fig. 9 are also due to inability of the heuristics and tree search of the branch-and-cut solver to consistently produce high-quality points (in terms of the proxy objective function), due to the hardness of this kind of combinatorial optimization problems. Furthermore, we think that the sharp improvements of the true objective function (AEP) in the mentioned figures show MILP's added value (and disprove the random search argument). For example, in Case I, when $N = 590$ and $K = 6$ the incumbent is improved more than $1\%$ in less than 20 minutes when four series of MILP runs are solved to optimality. With the number of possible combinations of new solutions for a given incumbent being at least $C_3^{574}$ (31'355,324), we strongly

believe that it would require significant luck for a random search to get quality solutions sampling from a pool of over 31 millions of alternatives in such a short time.

In conclusion, we see that, in general, the correlation between AEP and total wind speed deficit proxy persists.

- MILP modelling contribution: In the introduction (from lines 78 to 87) the works (Turner et al., 2014; Kuo et al., 2016) had been cited, referring to modelling WFLO in integer programming using approximated objective functions. To clearly indicate the positioning of our work, the following are added in the revised manuscript:

  "…This linearization approach is similar to (Turner et al., 2014…" (line 245)

  "…Compared to (Turner et al., 2014), the MILP program (23) with objective replaced by Eq. (24), linearizes the complexity of its largest set of constraints and variables from $N^2$ to $N$ (Eq. (23b) and Eq. (23e)). Furthermore, the constraints in Eq. (23d), which can lead to infeasible points, are not neglected as in (Turner et al., 2014) …" (line 272 to 274).

- Heuristic algorithm contribution: The use of heuristics is common in the field of integer programming. We had cited the works of (Fischetti et al., 2016; Shaw, 1998) and (Fischetti and Lodi, 2003) as a reference for local branching and neighborhood search theory. The main contribution claimed here is that the presented NSH, that limits the number of changes over binary variables $\xi$, using solution pool, and exploiting the strong correlation between a proxy (linear) objective function and a true (non-linear) objective function, is novel in the context of WFLO. The NSH is a specific purpose heuristic developed to address the WFLO problem that differs from any of the cited works and the references pointed out by the reviewer. It is hard to see, in our opinion, relevant resemblances with the Random Search of Feng et al. (randomly modifies the position of a single wind turbine at each iteration) and with the Bionic optimization of Song et al. (like the random search). The NSH relies on solving a sequence of MILP models using branch-and-cut (inner heuristics and tree search) to obtain high-quality solutions in terms of an objective function which approximates a true objective to optimize. This allows modifying multiple wind turbines simultaneously for analyzing possible layouts. The DEBO algorithm from *https://wes.copernicus.org/preprints/wes-2022-90* is discussed in the revised version of our manuscript in Introduction and Section 4. The comparisons are highlighted in Red color as this was a suggestion from the other reviewer.

  We have added this work in the introduction (paragraph from lines 99 to 109) as "The second main contribution is the proposition of a new special purpose neighborhood search heuristics in order to speed up the generation of high-quality solutions. This heuristic, wrapping both formulations, has a twofold functionality; first to increase tractability, and second to redirect the optimization search in terms of a specified objective function with higher fidelity. Similar neighborhood search methods have been proposed in the literature, as the Discrete exploration-based optimization (DEBO) (Thomas et al., 2022c), which is a two-steps process composed by a greedy initialization and a local search block. While the method proposed in this manuscript shares most of the advantages of the mentioned approach (no gradients required, can handle unconnected and non-convex boundary constraints, and so on), it actually improves the DEBO algorithm as among others, i) significantly less AEP function evaluations are required, and ii) it is based on well-establish integer programming

theory, relying in efficient implementations of the branch-and-cut algorithm. The main numerical results indicate good computational performances for a set of publicly available benchmark case studies compared to state-of-the-art gradient-free and gradient-based approaches (Baker et al., 2019)."

This article is referred back in this section in paragraph from lines 305 to 307 as

"… One of the advantages of the NSH compared to the DEBO algorithm (Thomas et al., 2022c) is the reduced number of AEP evaluations. In an iteration κ, only |S| evaluations are required. Likewise, many of the other expensive calculations are done in a pre-processing stage…"

"…Another difference between the NSH and the DEBO is that the latter only changes the position of a single WT in a given iteration, while the former considers simultaneous modifications of several WT positions." (lines 312 to 314).

- Computing time reference: Thanks a lot for this feedback. We agree with the reviewer, therefore the statements discussing computing time have been rephrased. The comparison of the NSH with gradient-free methods, including the pseudo-gradient, was rephrased, since the aim was to compare to typical metaheuristics that do not use at all gradients information, as genetic algorithm or swarm optimization, and not to the specific method. Lines 433 to 435 have been modified as
"…When directly comparing to typical metaheuristics (genetic algorithm, particles swarm optimization, etc), that do not use explicit gradients information, the presented method seems to perform well, being able of determining a similar layout quality in less than 2 h, which is generally faster than average computing time of these kind of algorithms…"
And lines 463 to 464:
"…which is generally a reasonable computing time compared to methods where gradients are not explicitly utilized in the optimization process, especially to metaheuristics as genetic algorithm or swarm optimization…"

As a general clarification regarding the computing time of the NSH in Fig. 5, Fig. 7 and Fig. 9, the following paragraph has been modified in the revised manuscript between lines 406 and 412:

"The NSH computing time results in Fig. 5 (blue line) do not reflect the instant where the incumbent is found, but the time progress of this algorithm, which is dependent on the execution of the MILP solver at each iteration. Table 2 contains information about the values of N, K, T, and termination criterion of the solver after each iteration κ of the NSH
Algorithm 1 (beginning from point 2 where κ = 1). This means that in iterations where the termination criterion is time (and not optimality), one could fine-tune T for an earlier stop, shortening the total time. This is particularly more relevant in cases where internal heuristics of the solver are activated at the root node of the search tree, coming up with the largest portion of solutions very early in the

process. Consequently, the total computing time, for all cases, is conservative and should be taken as an approximated reference."

This paragraph states the real meaning of computing time for the NSH in these figures. We have chosen to show them focusing on trends and potential rather than aiming at exhaustively comparing to all algorithms testbed that use different modelling philosophy and are designed for different purposes.
* * *
*CONNECTION OF KEY POINTS WITH SUBMISSION RATING*
* * *
*- Comment: By themselves, the application of a big-M trick to get a more efficient linearization and the design of the NSH are relevant contributions.*

*That, after the initial part of the optimization runs, more is happening than just a random search, is something that should be demonstrated, because otherwise I feel these contributions are not novel and proven enough to merit acceptance of the paper. The huge computational resources necessary for the piecewise-constant power curve linearization approach leads me to conclude that this approach is impractical, so effectively, that for that approach a negative result has been obtained. That is something that can be reported, but is by itself not enough to merit acceptance.*

*- Response:* We have elaborated in this response letter about the operating principle of the NSH and how this is beyond a random search of a single wind turbine in a given iteration. The main contribution stems from combining the linearization techniques within each MILP model, and the proposition of the NSH wrapping them. For example, the MILP model in Eq. (23) + objective Eq. (24) when solved as a full extensive model performs worse (in terms of final solution quality) than when this is embedded in the NSH (Fig. 5, Fig. 7, and Fig. 9). Conversely, the MILP model Eq. (23) + objective Eq. (24) enhances MILP model Eq. (18) for the fixed WT number case. While we agree that the computational resources are large, we see a great potential in applying them for hard WFLO instantiations, optimization objectives beyond AEP needs, or with complex site-dependent cost functions. Unified optimization with electrical systems would also benefit from this modelling technique. By comparing the results to a large testbed of algorithms, we aim to a proof of concept of the proposed methods, encompassing the MILP models and heuristic, rather than claiming computational superiority against the benchmark studies.
* * *
*DETAILED COMMENTS AND TECHNICAL FEEDBACK*
* * *
*Abstract*
* * *
*- Comment: velocity deficit → wind speed deficit*

*- Response:* Thanks for noticing this imprecision. It has been corrected as suggested.
* * *
*1: Introduction*
* * *
*- Comment: It is relatively long, so add subsections to help readers see its structured.*

*- Response:* Thanks for the valuable suggestion. We have added subsections within the introduction section as follows: 1.1 Motivation and Problem Definition, 1.2 Optimization Workflow for WFLO, 1.3 Continuous Optimization for WFLO, 1.4 Discrete Optimization for WFLO, 1.5 Literature Review for Integer Programming within WFLO, 1.6 Contributions.
* * *
- *Comment*: *l17: turns into a critical task → remains relevant (avoid exaggeration)*

- *Response:* This sentence has been deleted in the new version of the manuscript following a suggestion from the other reviewer. It reads as follows:

"…For wind energy to become the cornerstone of a successful green energy transition, further reduction in costs - partly achievable by economically refined wind farm designs - will play an important role."
* * *
- *Comment*: *multiple locations: Authors (Authors, year) → Authors (year) [just use \citet{…}?]*

- *Response:* Thanks for noticing this redundancy. We have fixed it accordingly.
* * *
- *Comment: l78-89: use list (first, second) to make this paragraph clearer.*

- *Response:* Thanks for the suggestion. In the new subsection 1.6 Contributions this list has been elaborated.
* * *
- *Comment: l92: Preface with, e.g., "The rest of the paper is structured as follows." for clarity.*

- *Response:* Thanks for the suggestion. We have fixed it accordingly.
* * *
- *Comment: l93: unfolds → describes? [strange wording]*

- *Response:* Thanks for the suggestion. We have changed this word as recommended.
* * *
- *Comment: l94: deployed → presented? [strange wording, occurring multiple times]*

- *Response:* Thanks for the suggestion. This work has been replaced by "shown", "depicted", and "introduced" in all the instances where they appeared.
* * *
*2: Physics Modelling*
* * *
- *Comment*: *l106: $\delta u\_il \to \delta\_il$: for mathematical notation, it is discouraged to use two (multiple) consecutive symbols, as this can be mistaken for the multiplication of two symbols; it also makes the notation heavier than needed.*

- *Response:* Thanks for the suggestion. We have changed this mathematical notation as recommended.
* * *
- *Comment: l107: mention that it is Case Study 1.*

- *Response:* Done.
* * *
- *Comment*: l109-110: inappropriate paragraph indent after math; avoid by not having blank line? [occurs very often; fix all].

- *Response*: Done. This type of inaccuracy have been fixed throughout the manuscript.
* * *
- *Comment*: l115: $\Delta u\_il \rightarrow \Delta\_il$ (as for $\delta u\_il$)

- *Response*: Thanks for the suggestion. We have changed this mathematical notation as recommended.
* * *
- *Comment*: l115: 'wind speed k': the use of k for a wind speed is confusing, as k is typically used for integers, but likely you meant k to be the index for a discretized wind speed value (then make that clear).

- *Response*: Done. As the reviewer suspects this is referring to a specific index for wind speed in a discretized wind rose. The sentence has been corrected as follows
"…and wind speed index k can be…"
* * *
- *Comment*: l124: drop 'and wind speed k'

- *Response*: Done
* * *
- *Comment*: l124-125: there is strange extra white space before the displayed equation; check your LaTeX [occurs very often; fix all].

- *Response*: Thanks for noticing this imprecision. We have fixed it accordingly throughout the manuscript.
* * *
- *Comment*: Eq. 6 is the logical combination rule when one wants a linear objective, but the one of Eq. 7 is used; this needs to be better justified.

- *Response*: Both superposition models are used. Eq. 6 is used for the power-curve-based model, and while it is also applicable to the power-curve-free model, we have chosen to use Eq. 7 since this optimization model is directly compared to the benchmark study in (Baker et al., 2019), which follows the root sum squares methodology. Thus, results in Sections 5.1, 5.2, 5.3, and 5.4 calculate AEP through Eq. 7 and Section 5.5 uses Eq. 6. Lines 234 to 236 explain this, while repeated in the introduction of Section V, paragraph from lines 320 to 326.
* * *
- *Comment*: l135: only the most simple models of power curves are convex right below rated speed.

- *Response*: True. An improved description of the power curve has been introduced in the revised manuscript.
* * *
*- Comment*: *l137: ≤ → < for middle two lines' left-hand inequality*

*- Response*: Thanks for noticing this typo. It had been fixed in advance.
* * *
*- Comment*: *Eq. 9: note that putting the sum over i after the w_jk corresponds to a more efficient calculation*

*- Response*: True. Please consider that this is just a representation in the text and does not reflect the actual code implementation.
* * *
*3: Optimization Models*
* * *
*- Comment*: *l153: '(2D in this study)': mention that when presenting the actual test case, not here.*

*- Response*: Done.
* * *
*- Comment*:  *l152: use of j different from wind direction index, so confusing double use of the same symbol; please avoid that.*

*- Response*: Thanks for noticing this typo. It has been corrected through the whole manuscript by replacing index $j$ to $q$ used in the distance functions.
* * *
*- Comment*:  *l164: what is meant by 'isometric'? [strange wording]*

*- Response*: In this context is used to describe the property of intervals splitting the cubic part of the power curve such as they all have the same length.
* * *
*- Comment*:  *Fig. 1: gray dot hard to see; please improve readability*

*- Response*: Done. Colors between the original power curve and the stepwise linear have been swapped.
* * *
*- Comment*:  *Eqs. 16 and 21 would be easier to understand if you first present the set same of equations for the non-linearized way in a grouped way, so that it is easy to see what changes; especially 16f is hard to understand without piecing together some things.*

*- Response*: Thanks for the feedback. The sequence of equations describing the characteristics of the sampling method applied to linearize the WT power curve has been extended, in an attempt to improve readability. We do feel though that the wake models inside the MILP formulations are adequately represented. Please note that from lines 223 to 229 the wake model is connected to Eq. (3) and Eq. (6) that in turn expresses the logic process to compute total wind speed at a given location.
* * *
*- Comment:  Eq. (16b): wouldn't ξ_i+sum_{j in N_i}ξ_j≤1 for all i be a more efficient set of constraints?*

*- Response*: Unfortunately, not. Note that in ξ_i+sum_{j in N_i}ξ_j≤1 if a ξ_j=1, then all ξ_l such as l in N_i and j ≠l will be forced to be zero, but not necessarily the condition d_{jl}<d^min would be satisfied, because the set N_i contains elements satisfying this
condition with respect to i. Therefore, it is mandatory that this constraint is N^2 in worst-case.
* * *
*- Comment:   l205: necessitated → needed*

*- Response*: Thanks, done.
* * *
*- Comment:   l206: total wind speed → sum of wind speeds ('total' was confusing to me here)*

*- Response*: Done.
* * *
*- Comment: l211: 'dropping the square roots' is a very crude way of introducing the linearization of the square root here; explain why you use this expression and not one with a different coefficient (you could also just put an abstract coefficient in front and drop it when you drop the first term).*

*- Response:* We acknowledge that simply dropping the square roots the model is not "correct", but expect that the resultant expression, incorporated in the MILP model, is "good enough".
Line 244 is added:
"…the arguments of the square roots in Eq. (19) define a function closely related to the full root-squared expression…"
To have a better idea about this premise, please see the below plot

[Figure]

The plane plot is of the function $z = x + y$ and the other one is for $z = \sqrt{x} + \sqrt{y}$. Note how these two functions follow relatively close to each other for non-negative values of $x$ and $y$. A

similar behavior is expected between the original root-squared expression and the one that ignores them. Practical evidence of the accuracy of this simplification is presented in Table 1.
* * *
- *Comment: l221: provide reference for big-M trick*

*Response:* Done.
* * *
*4: Neighborhood Search Heuristic*
* * *
- *Comment:  Alg.1 line 11: This is the first time 'solution pool' appears; this concept needs to be introduced on beforehand and it needs to explained concretely which solutions are in this pool and how they are generated (this is for me the part of the explanation that endangers reproducibility the most for me).*

- *Response:* Thanks for pointing this out. The following explanation of the solution pool has been added in lines 300 to 302.

"After solver termination, the solution pool S is retrieved in line 11. The solution pool contains all the feasible layouts obtained in an iteration κ from the MILP solver. These points are a result of a linear programming relaxation or from applying heuristics in a given node, such as, relaxation induced search, polishing, and feasibility pump (IBM, 2022)…"
* * *
*5: Computational Experiments*
* * *
- *Comment:  l288: V used in a second meaning from in Alg.1; avoid reusing mathematical symbols.*

- *Response:* Thanks for pointing out this redundancy. V for the number of wind speeds has been replaced by \Upsilon.
* * *
- *Comment: l330: relation between U and Ũ: not very interesting, rather between AEP and Ũ would be.*

- *Response:* This is exactly presented in Fig. 4c but instead of Ũ the component linked to deficit is plotted. A graph between AEP and Ũ would simply have a positive slope with same correlation.
* * *
- *Comment: l363: 'The inputs are tuned …': please share how this was done and how much effort this entails.*

- *Response:* This discussion has been extended in lines 392 to 405. The framework seems robust for wide internals of tunned values of these parameters, the presented here were obtained after few trial and error attempts.
* * *
- *Comment: l365: seek → search*

*- Response:* Thanks for the advice. Done.
* * *
*- Comment*: *The use of % in the discussion of 5.2-4 is confusing; a better approach is to use wake loss factor/percentage (1-AEP/AEP_wakeless) and mention percentage point changes; wake loss factor (differences) are independent of the starting layout and provide the scale of things that are of interest to wind farm developers (a 1 percentage point difference in wake loss factor is practically significant, but a 0.1 percentage point difference likely not anymore, as it will likely be below the uncertainty bounds*
*involved due to model uncertainty); wake loss factors can also be compared across wind farms, as they do not depend on the number of turbines.*

*- Response:* We do not really see a big added value by transforming the presented percentages from ((AEP_step/AEP_lastcommentedstep)-1) to (1-AEP_step/AEP_wakeless). The choice of showing this percentage calculation is to put it in the same context as the performance plots in Fig. 5, Fig. 7, and Fig. 9, which are presenting AEP vs computing time, while evaluating the improvement over a time slot.
* * *
*- Comment*: *There are numbers with 8 significant digits listed (some AEP values), which is absurd, as modelling inaccuracies make such precision unrealistic; try to give all numbers with a reasonable number of significant digits (giving more than 3 or 4 is typically dubious).*

*- Response:* We have corrected these numbers accordingly.
* * *
*- Comment*: *Figs. 5, 7, 9: put yellow box material in table for better legibility.*

*- Response:* Done.
* * *
*- Comment: l385: escalates → scales.*

*- Response:* Done.
* * *
*- Comment: l420-422: regarding the pattern in NSH operation: you cannot conclude what you have stated, as you always use smaller neighborhoods before larger ones; making statements like this would require comparison at least with runs where the neighborhood size decreases; I would just leave out this statements.*

*- Response:* Done.
* * *
*- Comment*: *l451: you state that the power curve-free approach is not suited for NPV optimization, but as part of the linearization, it is entirely possible to estimate a coefficient to transform (m/s)² to units of currency; so I find the current argumentation to a be a bit limited; I'd suggest including a more nuanced argumentation.*

*- Response:* We are very aware of this possibility. Such a linearization would require performing a single variable linear regression mapping wind speed to units of currency. It is very important in this approach to properly weight both cost components (cost of wind turbines and energy revenues) for the optimization not to be ill-conditioned. We have elaborated the following correlation plot for a case of variable number of wind turbines (ranging from 10 to 100) using the power-curve-free model:

[Figure]

From the plot, one can conclude that in some cases there could be a good correlation between AEP and total wind speed deficit, however in other cases the correlation is quite poor. If one "hits" the case situate on the left side of the plot, the mapping could give good results, however this would be rather random. The following statement has been added between lines 486 and 490.

"…Even if the power-curve-free model (Sect. 3.2) exhibits a quite good performance in terms of AEP and computing time for fixed number of WTs (when AEP and NPV are basically the same metric), it is not very well suited when variable number of wind turbines are considered. Based on computational experiments not included in the paper, the power-curve-free model embedded in the NSH terminates too early in the search process, resulting in a worse solution than the alternative discussed in the following…"
* * *
*- Comment*: *l463: 'Mill.Eur' is not a proper unit; please use proper units also for currencies.*

*- Response:* We have chosen to replace 'Mill. Eur' by m EUR according to https://publications.europa.eu/code/en/en-370303.htm#:~:text=10%20bn%20GBP-,NB%3A,is%20insufficient%20for%20spelling%20out.

[Figure]
* * *
*6: Discussion*
* * *
*- Comment*: l499-500: Is it hard to establish whether a MILP formulation would still be possible? As long as the wake model enters into the picture via pairwise deficits, it should work without material changes.

*- Response:* We refer to the fact that the thrust coefficient is function of the local turbine wind speed, which in turn affects the pairwise deficits. In a MILP formulation these deficits are computed in advance if the program is extensively formulated as a fully compact model. Then, it is not possible to calculate them in function of the local speed before knowing the positioning of the wind turbines. Strategies as lazy constraint callbacks may be handy for this issue, but we have not done the corresponding rigorous analysis to determine with certainty at what extent this is accurate.
* * *
*- Comment*: l507-508: Why hasn't the study for the dependence on initial layout been done yet? As far as I understand, it is mostly just a matter of rerunning the cases n times. (While I understand that this takes time, the optimization runs would not require full convergence, but just long enough to see repeated/varying behavior.) This would be an interesting addition to the paper and increase its value.

*- Response:* We place this analysis as an extra study to be formally addressed and presented in subsequent research, as the emphasis of this article has been into describing the proposed integer programming models and the heuristics. We do not expect that by changing initial layouts, the final solution will be markedly changed and typically linear integer programming models are not as dependent to the initial layout as gradient-based methods. Considering the actual article length (32 pages), this extra content may not get sufficient attention, as other very important points as both integer linear programs, correlation analysis, NSH operation, global model performance analysis, and non-classic WFLO have been prioritized. To avoid confusions, we have removed the future work part from the manuscript.
* * *
We strongly believe that the manuscript has gained strength, consistency, coherence, and formality by the changes suggested by the Reviewers. We thank them and look forward to hearing from them again.

Best wishes,

[revised manuscript text omitted]

---

## Referee Report (RR1)

**General comments**

The paper has been significantly improved and the authors have made good efforts to address the reviewers concerns.

There are still several places where the paper has unusual English usage. There are still a few typos as well. I recommend that the authors work through the paper with a native English speaker and editor to address the typos and unusual language usage.

**Detailed comments**

I am still not completely satisfied with the explanation and discussion around the linearization applied by dropping the square root in Eq. 20 of the revised manuscript. I would like to see more justification for this approach. I understand the approach may be acceptable because the term within the square root will generally be between 0 and 1, where the sum and the square root of the sum are similar. However, the authors do not give such an explanation in the manuscript. I also think it would be useful to recognize that dropping the square root may introduce as much as 25% error to the terms inside the square root depending on the wake deficits in the farm. The 25% error may be acceptable in this case, and I expect the error will usually be closer to 10%, but I think it bears mentioning.

---

## Author Response (AR2)

DTU Wind

[Figure]

Wind Energy Science

1st August 2023

Prof. Michael Muskulus
Associate Editor of the Journal,
Editorial Board,
Reviewers

**RESPONSE TO THE EDITOR AND REVIEWERS**

Dear Editor and Reviewers,

We would like to thank you all for your effort on reviewing our article entitled *"A Neighborhood Search Integer Programming Approach for Wind Farm Layout Optimization"* (submission wes-2022-82). It is very satisfying to us that you consider that the manuscript is almost ready to be published.

To finally fulfill the last requirements, we have addressed point by point the comments from the reviewers after revision 1. We would like to emphasize our appreciation to the reviewers for the very high quality of their comments, resulting in a considerable improvement of the manuscript. See below the responses.
* * *
*- Comment: p2l33, p5l124: the Bastankhah's Gaussian → Bastankhah's Gaussian*

*- Response:* Thanks for the suggestion. We have taken it into account.
* * *
*- Comment: p2l46-47: try to find and add a reference that supports your claim of poor scalability or at least make it more nuanced; there are also gradient free algorithms that scale well (as mentioned in my original report)*

*- Response:* We agree with the reviewer about the need to be more specific in this statement. One could classify gradient-free methods in two subcategories: heuristic and metaheuristics. Heuristics usually scale well (polynomially), while metaheuristics (such as population-based) usually exhibit exponential complexity. We have added the reference (Stanley and Ning, 2019) where there is available a quantitative analysis of gradient-based and gradient-free methods scalability, justifying our claim. The sentence now reads as follows:
*"In general, metaheuristic algorithms, although highly flexible for modelling aspects, have considerably poorer scalability for larger problem sizes than gradient-based approaches (Stanley and Ning, 2019)."*
* * *
*- Comment: p3l70: try to find and add a reference that supports your claim of continuous-location algorithms not supporting NPV optimization*

*- Response:* We have added reference (Pollini, 2022) where the point of number of wind turbines as variable is discussed along with an extensive literature review.
* * *
REG-no. DK 30 06 09 46

Technical University of Denmark
**Department of Wind and Energy Systems**

Frederiksborgvej 399
Building 118
4000 Roskilde
Denmark

Tel  +45 93 51 89 91
Dir.  +45 93 51 89 91

juru@dtu.dk
www.vindenergi.dtu.dk

*- Comment: p3l84: try to find and add a reference that supports your claim of heuristic routines not being compatible with continuous-location optimization; wake width expansion can be seen as such a heuristic, but also your NSH could in principle be compatible with an inner continuous-location optimization*

*- Response:*  We refer in this sentence to heuristics following a strict definition of these algorithms in the context of integer programming theory, widely applied in routing problems, for example. Meaning that integer variables are restricted to a smaller subdomain to prove local optimality. This claim is also backed by references  (Fischetti et al., 2016; Shaw, 1998).
* * *
*- Comment:* p4l93: 'previous works' → cite (some of) them

*- Response:*  In subsection 1.5 this point is discussed. To avoid redundancy, the sentence has been modified to

"*For example, as discussed before, previous works have considered aggregation of power deficits instead of velocities, gaining a simplification on the mathematical formulation in detriment of the physics modelling fidelity.*"
* * *
*- Comment:* p4l96 and in many other instances: avoid citations that break the flow when reading out loud, but make citations part of the sentence; here 'In contrast to (LoCascio et al. 2022)' → 'In contrast to LoCascio et al. (2022)'

*- Response:*   Thanks for the suggestion. We have taken it into account throughout the whole manuscript.
* * *
*- Comment:* p5l145: collects → collect

*- Response:*   Done.
* * *
*- Comment:* p6l147: the → The

*- Response:*   Done.
* * *
*- Comment:*  p7l182: scrap first $u\_ijk$?

*- Response:*   We do not understand what is meant by this comment.
* * *
*- Comment:*  p7l185: why not use 0 instead of some $u^{ini}$

*- Response:*   We feel comfortable with this explicit notation, and we do not think it affects the formality of the model deployment.
* * *
*- Comment:* p7l189-196: join equations in one nice align or gather, as the current display has too much whitespace in between, interrupting the reading flow

*- Response:* Thanks for the suggestion. We have taken it into account.
* * *
*- Comment:* p8l202: Let → Let us

*- Response:* Since we chose to use impersonal language in the manuscript, we have edited this sentence as:

"Let binary state variables $\eta_{lijk} \in \{0, 1\}$ for $l = 1, \ldots, m + 2$ be defined with the interpretation"
* * *
*- Comment*: p9l228: Eq. (18f) → Eq. (18g)?

*- Response:* Done.
* * *
*- Comment*: p9l232-233: neglects power curve → neglects the power curve

*- Response:* Done.
* * *
*- Comment*: p9l234: 'The model' → Which?

*- Response:* Clarified.

"The power-curve-free model introduces…"
* * *
*- Comment*: p10l252: ,z_ij-ξ_i,z_ij-ξ_j → I am a bit confused by these math fragments

*- Response:* Thanks for noticing this typo. The correct fragments are $z_{ij} \leq \xi_i$, $z_{ij} \leq \xi_j$, $z_{ij} \geq 0$.
* * *
*- Comment*: p10l254,p11(23b): explain the big-M trick a bit

*- Response:* We consider the explanation suffices in this case.
* * *
*- Comment*: p11l274: points → solutions(?)

*- Response:* We consider points as elements forming the domain, while solution being the optimum or incumbent evolution. Hence, we keep this use throughout the manuscript.
* * *
*- Comment*: straight forward → straightforward

*- Response:* Done.
* * *
*- Comment*: p11l281-282: what is meant by 'determined' (clarify/reformulate)

*- Response:* Done.

"…for the power-curve-based model, the value of continuous state variables u can be found through classical wake analysis…"
* * *
*- Comment*: p12Alg1l12: on first mention of 'true objective function', be sure to be explicit what is meant by this

*- Response:* Done. This is explained in the algorithm description p11l289 to l293.
* * *
*- Comment*: p14l328: Wind resource → The wind resource

*- Response:* Done.
* * *
*- Comment*: p15l357-358: Describe the generation procedure more explicitly; I disagree that the general trend is representative, as you nicely explain in your replies to reviewer comments (I had hoped some of that explanation would have found its way into the paper)

*- Response:* To nuance this statement, the sentence has been modified to
"…Although the random way of generating the layouts is biased against high-quality points, the interest is in the general trend in order to assess whether it makes sense to implement the linear proxy objective $\sum_{i=1}^{N} \tau_i$ when optimizing AEP…"
This is aligned with the discussions during the reviewing process. We decided to not extend the content of the manuscript with new results obtained after replying to the reviewers' comments, since we consider that the presented findings in the article are enough to support the main results while avoiding excessive length. We think that by making open access to the response letter, we are explicitly being transparent about detailed information of the methods' performance.
* * *
*- Comment*: p17l391 and further subsections: repeat the case being discussed in-text for better text flow

*- Response:* Done.
* * *
*- Comment*: p19Table2 and further such tables: put into an appendix, as it doesn't add much beyond what is in the figures and does take up space/interrupts the text

*- Response:* Done.
* * *
*- Comment*: p19l435-438: try to find and add a reference that supports your claims of 'generally faster' and 'cannot support', or be more nuanced

*- Response:* The expression is modified as per
"…which is usually a competitive time compared to these kinds of population-based algorithms."
* * *
*- Comment*: p21l442: we present → present

*- Response:* Done.
* * *
*- Comment*: p26l490: in the following → below

*- Response:* Done.
* * *
*- Comment*: p26l508: as target → as the target

*- Response:* Done.
* * *
*- Comment*: * p28l544-549: use of a much finer wind rose is another important aspect (cf. Thomas 2022c)

*- Response:* True. Although the power-curve-free model in principle should not be directly affected by it in terms of model scalability.
* * *
*- Comment*: p28l550:very enthralling → strange formulation

*- Response:* We removed the "very".
* * *
*- Comment*: 28l553: very challenging → I find this claim to be insufficiently justified (e.g., by references or argumentation)

*- Response:* We consider that we provide key references and raise challenges not addressed in the literature (Sect. 1.4). Regarding the complexity of having the number of wind turbines as variable in the context of continuous gradient-based optimization, we cite the reference (Pollini, 2022). For the integrated optimization of wind turbines and cable layout, we have added references (Pérez-Rúa and Cutululis, 2022; Cazzaro et al., 2023), where this challenge is discussed. For the ease of modelling terrain/local costs, reference (Cazzaro and Pisinger, 2022) is incorporated. On the ease of modelling different project areas and several discretized wind turbine types, we consider that these advantages are intuitively understandable.
* * *
*- Comment*: p29l570: is inherently studied → strange formulation

*- Response:* We removed "inherently".
* * *
*- Comment*: p29l571: I would say that the AEP is actually comparable even

*- Response:* We agree.

Best wishes,
Juan-Andrés Pérez-Rúa
Mathias Stolpe
Nicolaos A. Cutululis